# Identifying Robust Neural Pathways: Few-Shot Adversarial Mask Tuning for Vision-Language Models

**Wonjeong Choi**[1], **Sejong Ryu**[1], **Jungmoon Lee**[1], **Dong-Jun Han**[2], **Jaekyun Moon**[1]

[1] Korea Advanced Institute of Science and Technology (KAIST),

[2] Yonsei University

[1] {dnjswjd5457,ryusejong,wndanseh}@kaist.ac.kr;

[2] djh@yonsei.ac.kr; [1] jmoon@kaist.edu

## Abstract

Recent vision-language models (VLMs), such as CLIP, have demonstrated remarkable transferability across a wide range of downstream tasks by effectively leveraging the joint text–image embedding space, even with only a few data samples. Despite their impressive performance, these models remain vulnerable to adversarial attacks, raising significant concerns about their security and reliability in practical deployments. To address this issue, we propose Adversarial Mask Tuning (AdvMask), a method that effectively enhances the robustness of VLMs without directly modifying their pre-trained weights. Instead, our AdvMask learns a set of binary masks that selectively deactivate model parameters vulnerable to adversarial perturbations. By identifying robust neural pathways within the vision encoder, AdvMask facilitates the generation of features and predictions that are resistant to adversarial attacks. Furthermore, we introduce a Layer-wise Adaptive Feature Alignment (LAFA) loss, specifically designed to optimize AdvMask in few-shot scenarios. The LAFA loss adaptively aligns intermediate-layer features from clean and adversarial samples across each transformer block, enhancing the representational robustness of the model. Experimental results across multiple benchmarks confirm that our AdvMask approach substantially outperforms existing adversarial tuning techniques for VLMs, especially in few-shot settings. Our code is available in https://github.com/wonjeongchoi/AdvMask.

## 1 Introduction

Vision-Language Models (VLMs), such as CLIP (Radford et al., 2021), have demonstrated exceptional zero-shot generalization capabilities and impressive transferability across a wide range of downstream tasks, gaining significant attention in recent years (Zhang et al., 2024b). By bridging the semantic gap between visual and textual representations through contrastive learning, they have enabled high-level understanding and versatile potential across various applications.

**Motivation.** Despite significant advancements, VLMs remain vulnerable to adversarial attacks, which restricts their practical deployment in real-world downstream tasks. This inherent weakness significantly undermines their reliability and trustworthiness, raising concerns in safety-critical and security-sensitive downstream applications such as autonomous driving (Tuncali et al., 2018; Deng et al., 2020), medical analysis (Buch et al., 2018; Finlayson et al., 2019), and manufacturing systems (Picard et al., 2023). Consequently, there is a pressing need to develop algorithms that achieves robustness against adversarial perturbations during downstream tasks. This problem becomes even more pronounced in *few-shot settings* (Dong et al., 2022; Wang et al., 2020), where the number of training samples available for the downstream task is severely limited (e.g., medical applications).

**Challenges.** Recently, researchers have explored techniques to strengthen the adversarial robustness of VLMs (Zhao et al., 2023; Cui et al., 2024). Among these, adversarial tuning of textual or visual prompts (Zhou et al., 2024; Mao et al., 2023; Zhang et al., 2024a) has widely adopted as a prominent method, aiming to improve the model's predictive robustness by carefully modifying the prompts

to resist adversarial perturbations. While these approaches only require updating a small number of learnable parameters, they overlook the inherent properties in the model's pre-trained structure (i.e., neurons), limiting their capability to produce robust representations against adversarial attacks. Other works attempt to directly fine-tune the model using adversarial training strategies; however, these approaches can lead to overfitting in few-shot settings (where only a small number of labeled samples are available for each downstream task) and may compromise the generalization ability of the original pre-trained VLM. Furthermore, several methods targeting zero-shot robustness (Yu et al., 2024; Mao et al., 2023) rely on a held-out dataset for adversarial tuning (i.e., no task-specific samples are available), but they often fail to achieve satisfactory performance on downstream tasks. The effectiveness of these approaches largely depends on the quality of the held-out dataset. An extended discussion of related works is provided in Sec. 4. Motivated by these challenges, in this work, we aim to answer the following key question:

*What is the most effective way to achieve robustness against adversarial attacks on pre-trained VLMs in few-shot downstream settings?*

**Key Ideas.** Unlike previous methods that predominantly focus on prompt adaptation or direct parameter updates, we propose an *adversarial mask tuning* (AdvMask) approach that searches for robust subnetwork within well-trained VLMs as a promising alternative. Inspired by recent studies (Zheng et al., 2023; Zhao et al., 2020; Lin et al., 2020) demonstrating the effectiveness of identifying neural pathways for adapting large-scale pre-trained models, we introduce a novel perspective of a _robust neural pathway_, which, to the best of our knowledge, has not been explored in previous works. Specifically, given a few samples from the downstream task, our goal is to learn a binary mask that identifies a subnetwork structure within the pre-trained VLM, one that not only facilitates downstream adaptation but also inherently resists adversarial perturbations. Consequently, by identifying the robust neural pathway, our approach selectively emphasizes robust features during forward passes, substantially improving the adversarial robustness. Interestingly, we demonstrate that such a robust neural pathway indeed exists (further intuitive explanations are provided in Sec. 3.3).

Within our AdvMask training paradigm, we introduce the Layer-wise Adaptive Feature Alignment (LAFA) loss, which enables enhanced robustness and stability. Previous objective functions for adversarial tuning (Mao et al., 2023; Zhou et al., 2024) primarily provide supervision at the final output stage (i.e., the joint text-image embedding space), overlooking the importance of robust intermediate representations within the vision encoder. In contrast, our LAFA loss explicitly guides each transformer's intermediate representations to be robust against adversarial perturbations by closely aligning features extracted from adversarial samples with their corresponding clean sample features. Additionally, to effectively handle the limited data in few-shot settings, we adopt an adaptive weighting mechanism based on predictive reliability. Specifically, within our LAFA loss, features from samples that the model predicts correctly with high confidence provide more reliable alignment signals, whereas samples predicted with lower confidence contribute less, preventing unstable or incorrect optimization. This carefully designed LAFA loss encourages consistent intermediate feature representations between clean and adversarial inputs, improving adversarial robustness in our few-shot AdvMask framework.

**Summary of Contributions.** Overall, we introduce the notion of robust neural pathway and make the following key contributions:

- We propose a new few-shot Adversarial Mask Tuning (AdvMask) framework that effectively enhances the adversarial robustness of VLMs by identifying robust sub-network structures using binary masks, without modifying their pre-trained weights.

- We introduce a Layer-wise Adaptive Feature Alignment (LAFA) loss, specifically designed to optimize AdvMask training in few-shot scenarios. The LAFA loss adaptively aligns intermediate-layer features between clean and adversarial samples to find the robust neural pathway.

Experiments across various downstream datasets demonstrate that AdvMask consistently improves few-shot adversarial robustness over existing baselines. Moreover, since AdvMask learns and stores only binary masks corresponding to a subset of model parameters, it is highly parameter-efficient, reducing memory requirements during training and inference (see Appendix Sec. D.7 for details).

## 2 ADVMASK: FEW-SHOT ADVERSARIAL MASK TUNING FOR VLMS

### 2.1 PRELIMINARY AND PROBLEM SETUP

**CLIP Recap.** In this paper, following prior works on adversarial robustness of VLMs (Zhou et al., 2024; Mao et al., 2023; Yu et al., 2024), we mainly use the CLIP (Radford et al., 2021) as our target VLM. We also provide results on other VLM, VisualBERT (Li et al., 2019), in Sec. C.4 of Appendix. CLIP consists of an image encoder $I(\cdot)$ and a text encoder $T(\cdot)$, which project images and text into a joint embedding space via contrastive learning on large-scale paired datasets. This enables strong zero-shot classification performance on diverse image recognition tasks. For a downstream classification task with images $\{x_1, \ldots, x_m\}$ and labels $y \in \{1, \ldots, K\}$, each label $y_i$ is embedded into a textual prompt (e.g., "a photo of a [class]") to form input $t_i$, yielding a text representation $z_T(t_i)$. Similarly, an input image $x$ is encoded by the image encoder, typically implemented as a vision transformer (ViT) (Dosovitskiy et al., 2020), to produce $z_I(x)$. Finally, the probability that image $x$ belongs to class $y_i$ is calculated as:

$$p(y = i \mid x) = \frac{\exp(\cos(z_T(t_i), z_I(x))/\tau)}{\sum_{j=1}^{K} \exp(\cos(z_T(t_j), z_I(x))/\tau)}, \tag{1}$$

where $cos(\cdot, \cdot)$ denotes cosine similarity and $\tau$ is a learnable temperature parameter.

**Few-Shot Adversarial Tuning.** Given a pre-trained VLM with strong generalization capabilities, our goal is to adapt it for adversarial robustness in few-shot scenarios, where only 1–16 samples per class are available from the downstream dataset. In such settings, learning adversarial robustness is particularly challenging due to limited supervision. To address this, rather than relying on cost-intensive methods that fine-tune all parameters of large-scale VLMs, we adopt an efficient mask-tuning strategy. This approach keeps the pre-trained weights fixed while optimizing only binary masks over selected parameters, enabling the discovery of robust neural pathways.

### 2.2 MASK TUNING FOR ADVERSARIAL ROBUSTNESS

In this work, we propose AdvMask, a novel adversarial mask-tuning approach to enhance adversarial robustness. As illustrated in Fig. 1, AdvMask builds on recent mask-tuning techniques (Zhao et al., 2020; Zheng et al., 2023), which identify sub-networks within pre-trained models for improved adaptation. Specifically, by optimizing binary masks on the vision encoder's pre-trained parameters, AdvMask deactivates adversarially vulnerable weights and identifies robust neural pathways that yield stable, resilient visual representations, enabling reliable predictions under attack. To elaborate on our method, we first detail how the mask parameters can be optimized efficiently, and subsequently extend this to improve the adversarial robustness.

**Mask Tuning.** Given the pre-trained weights $\theta$ of the image encoder $I(\cdot)$, we first define a real-valued mask $M$ of the same size. A binary mask $M_{bin}$ is then obtained by thresholding with $\alpha$:

$$M_{bin} = \mathbb{I}[M > \alpha], \tag{2}$$

where $\mathbb{I}[\cdot]$ is an indicator function used for binarization. We compute the masked weights $\theta'$ through an element-wise product (i.e., Hadamard product, $\odot$) as $\theta' = \theta \odot M_{bin}$, and the encoder produces the visual representation $z_I(x; \theta')$ for input $x$. However, direct optimization of the binary mask $M_{bin}$ is infeasible due to non-differentiability of binarization function in Eq. 2. To overcome this, following previous works (Zhao et al., 2020; Lin et al., 2020), we employ the Straight-Through Estimator (STE) (Bengio et al., 2013), allowing indirect updates to the real-valued mask $M$ as:

$$M \leftarrow M - \gamma \cdot \frac{\partial L}{\partial M_{bin}}, \tag{3}$$

where $\gamma$ is the learning rate, and $L$ is the objective function for mask tuning.

**Adversarial Mask Tuning (AdvMask).** Beyond the downstream adaptation, we extend the mask tuning to adversarial robustness by optimizing binary masks that selectively deactivate parameters vulnerable to perturbations, enabling stable predictions. Given clean samples $x$ and adversarial counterparts $\tilde{x}$ with labels $y$ from a few-shot dataset $S$, our goal is to tune the binary mask $M_{bin}$

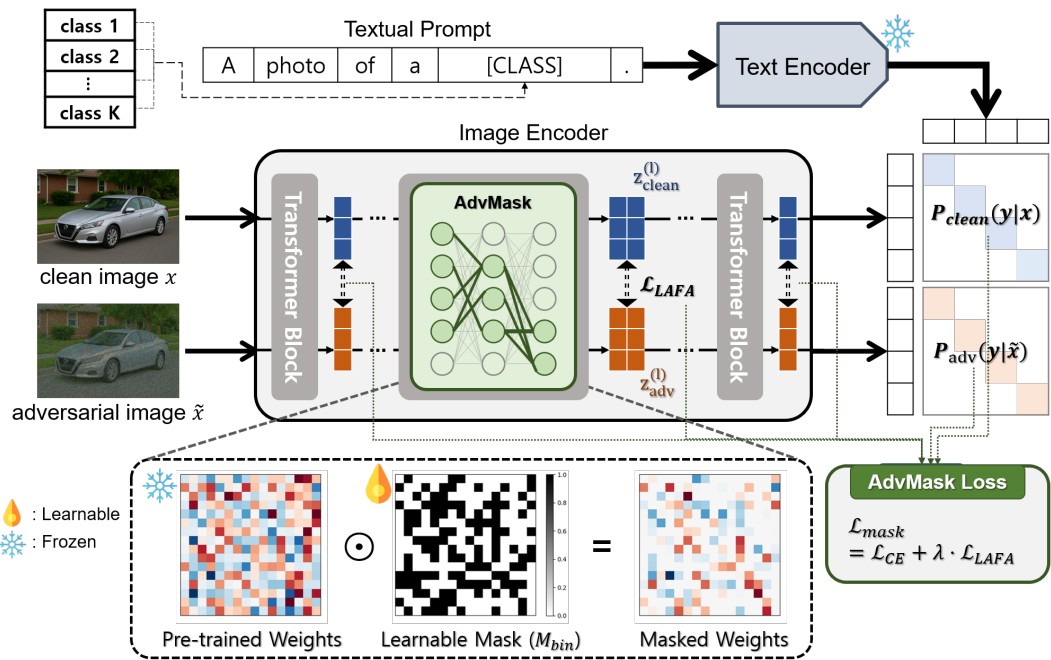

Figure 1: Overview of our AdvMask. Given clean and adversarial inputs, AdvMask learns binary masks $M_{bin}$ (shown as black-and-white grids) that selectively deactivate parameters vulnerable to adversarial perturbations. The masks enforce consistency between clean and adversarial intermediate representations ($z_{\text{clean}}^{(l)}$, $z_{\text{adv}}^{(l)}$) via our layer-wise adaptive feature alignment loss ($\mathcal{L}_{\text{LAFA}}$ in Sec. 2.3), combined with adversarial cross-entropy loss ($\mathcal{L}_{\text{CE}}$). This identifies robust neural pathways in the vision encoder without modifying pre-trained weights.

while keeping the pre-trained weights $\theta$ fixed. This adversarial tuning can be formulated as a min-max optimization problem, where the inner maximization step generates adversarial samples $\tilde{x}$ as:

$$\tilde{x} = \arg \max_{\tilde{x}:|\tilde{x}-x|\leq\epsilon} \mathcal{L}\left(f_{\theta\odot M_{bin}}(\tilde{x}, t), y\right), \tag{4}$$

where $\delta = \tilde{x} - x$ is the perturbation bounded by $\epsilon$, $\mathcal{L}$ is the loss function (e.g., cross-entropy loss) for generating adversarial perturbations, and $f_{\theta\odot M_{bin}}(\cdot, t)$ represents the model output given the binary mask $M_{bin}$ and textual prompts $t$. Subsequently, the outer minimization updates the mask parameters $M_{bin}$ to minimize the adversarial loss using the generated adversarial examples as:

$$\min_{M_{bin}} \mathbb{E}_{(x,y)\sim S}\left[\mathcal{L}_{\text{mask}}\left(x, \tilde{x}, t, M_{bin}, y\right)\right]. \tag{5}$$

By carefully designing the tuning objective ($\mathcal{L}_{\text{mask}}$) and learning the mask, our AdvMask identifies robust sub-networks without altering the pre-trained parameters. This enables significant robustness gains in few-shot scenarios while preserving the generalization capabilities of the fixed pre-trained model. In Sec. 2.3, we introduce our loss design tailored for challenging few-shot settings.

**Key Advantages of our AdvMask.** Our proposed AdvMask offers practical advantages for adversarial robustness. First, it adapts only a binary mask without altering pre-trained weights, preserving generalizable knowledge while regulating information flow to produce stable and robust representations on both clean and adversarial samples (further insights are provided in Sec. 3.3). Second, AdvMask effectively leverages limited few-shot data to selectively activate or deactivate crucial pathways, improving transferability to downstream tasks and significantly enhancing robustness against adversarial attacks. These advantages make AdvMask a parameter-efficient solution for strengthening VLM robustness across diverse real-world scenarios.

## 2.3 LAYER-WISE ADAPTIVE FEATURE ALIGNMENT (LAFA) LOSS

**Motivation.** Our goal is to tune a binary mask that enhances adversarial robustness in few-shot settings. Prior objectives for robustness (e.g., TeCoA (Mao et al., 2023)) mainly supervise the final

output space (i.e., the joint text–image embedding space), which limits their ability to enforce robust intermediate representations and provides insufficient learning signals under scarce data. By contrast, AdvMask adapts internal parameters of the image encoder, where robust intermediate features are crucial. To this end, we propose a layer-wise feature alignment loss applied across encoder blocks, explicitly promoting stable representations and providing stronger guidance for few-shot mask tuning.

**Loss Formulation.** To explicitly guide robust feature representations, we propose a Layer-wise Adaptive Feature Alignment (LAFA) loss. This loss aligns adversarial features (from perturbed inputs) with clean features at each transformer layer of the image encoder, encouraging stable and robust intermediate representations. The intuition is that small adversarial perturbations, though imperceptible at the input, can amplify through deeper layers. By learning binary masks that deactivate vulnerable parameters at each layer, AdvMask suppresses this propagation and promotes robustness. Since intermediate layers lack explicit label supervision, we leverage clean features as targets, aligning adversarial features to them during tuning. Formally, the loss is defined as:

$$\mathcal{L} = \frac{1}{|L| \cdot |\mathcal{B}|} \sum_{l \in L} \sum_{x \in \mathcal{B}} \| z_{\text{clean}}^{(l)} - z_{\text{adv}}^{(l)} \|_2^2, \tag{6}$$

where $\mathcal{B}$ is the sample batch, $L$ the set of layers for alignment, and $z_{\text{clean}}^{(l)}$ and $z_{\text{adv}}^{(l)}$ the output features at the $l$-th transformer layer of our masked image encoder for clean and adversarial samples, respectively.

**LAFA Loss.** To further elaborate the learning signals and ensure stable optimization in few-shot scenarios, we propose a Layer-wise Adaptive Feature Alignment (LAFA) loss with an adaptive weighting scheme based on predictive reliability. The key idea is that if the model fails to correctly predict a clean sample, its feature may serve as a noisy alignment target, which is especially harmful under data scarcity as it can mislead mask optimization in unintended or sub-optimal directions. To mitigate this, we weight each sample by its predictive reliability (i.e., confidence in the ground-truth class), enabling adversarial features to align more strongly with reliable clean features and less with unreliable ones. Formally, our LAFA loss is defined as:

$$\mathcal{L}_{\text{LAFA}} = \frac{1}{|L| \cdot |\mathcal{B}|} \sum_{l \in L} \sum_{x \in \mathcal{B}} \frac{p(y|x)}{\mathbb{E}_{\mathcal{B}}[p(y'|x')] + \epsilon} \| z_{\text{clean}}^{(l)} - z_{\text{adv}}^{(l)} \|_2^2, \tag{7}$$

where $p(y|x)$ is the masked model's confidence for the ground truth class $y$ given the clean input $x$, and the denominator (i.e., $\mathbb{E}_{x' \sim \mathcal{B}}[p(y'|x')] + \epsilon$) normalizes weights, with a small constant $\epsilon$ for numerical stability. As a result, our LAFA loss prioritizes samples with clear and informative representations during alignment, further improving robustness, particularly under few-shot scenarios.

**Final Objective.** Our final tuning objective for AdvMask combines the cross-entropy (CE) loss on adversarial samples $\tilde{x}$ with our proposed LAFA loss, optimizing a set of binary mask parameters without modifying the pre-trained weights as:

$$\mathcal{L}_{\text{mask}} = \mathcal{L}_{\text{CE}}(\tilde{x}, y) + \lambda \cdot \mathcal{L}_{\text{LAFA}}(x, \tilde{x}, y), \tag{8}$$

where $\lambda$ is a coefficient balancing the two losses. This combination of objectives complements each other in tuning our adversarially robust mask: while the adversarial CE loss directly enhances prediction-level robustness, our LAFA loss ensures robust intermediate representations by explicitly guiding the learned binary mask to generate consistent features for both clean and adversarial samples. Consequently, our carefully designed loss function significantly improves adversarial robustness, especially in challenging few-shot adversarial tuning scenarios.

## 3 EXPERIMENTS

### 3.1 EXPERIMENTAL SETTINGS

In this section, we conduct extensive experiments to demonstrate the effectiveness of our AdvMask approach for enhancing adversarial robustness. Basically, we follow the few-shot adversarial tuning setup from Zhou et al. (2024). Specifically, for adversarial tuning of the CLIP model with each baseline method, we randomly sample 1, 2, 4, 8, and 16-shot samples per class from the training set of each downstream dataset. We then evaluate the tuned model on the test dataset by measuring classification accuracy ($\%, \uparrow$) on clean samples and their adversarially perturbed samples separately.

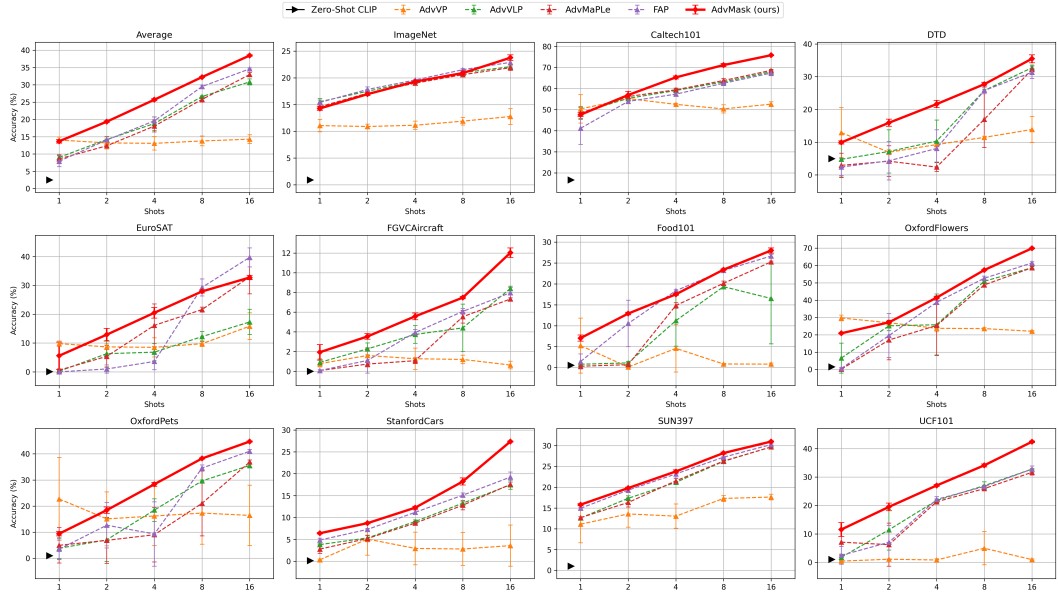

Figure 2: Adversarial test accuracy (%, ↑) over 11 datasets in few-shot settings. Results are averaged over 3 random trials (full results with standard deviations are provided in Sec. B.1 of Appendix).

**Datasets.** Following previous studies on adversarial robustness of CLIP models, we evaluate our AdvMask method across various image classification datasets. Specifically, we consider general object datasets (ImageNet (Deng et al., 2009), Caltech101 (Fei-Fei et al., 2004)), a texture recognition dataset (DTD (Cimpoi et al., 2014)), fine-grained object datasets (FGVCAircraft (Maji et al., 2013), OxfordPets (Parkhi et al., 2012), Flowers102 (Nilsback & Zisserman, 2008), Food101 (Bossard et al., 2014), and StanfordCars (Krause et al., 2013)), a scene recognition dataset (SUN397 (Xiao et al., 2010)), an action recognition dataset (UCF101 (Soomro et al., 2012)), and a satellite imagery dataset (EuroSAT (Helber et al., 2019)).

**Baselines.** To validate the effectiveness of AdvMask in realistic scenarios requiring efficient tuning for large-scale VLMs, we mainly compare our method with parameter-efficient adversarial prompt-tuning methods widely used in prior robustness studies (Zhou et al., 2024; Mao et al., 2023; Zhang et al., 2024a), as well as with TGA-ZSR (Yu et al., 2024), which fully adapts model parameters in zero-shot robustness experiments. Our adversarial prompt-tuning baselines include adversarial visual prompt tuning (AdvVP) (Mao et al., 2023) with hand-crafted textual supervision, adversarial visual-text prompt tuning (AdvVLP) with independently learnable prompts, adversarial multi-modal prompt learning (AdvMaPLe) (Khattak et al., 2023), and the recent few-shot adversarial prompt learning (FAP) (Zhou et al., 2024). We also report the compatibility of AdvMask with a learnable prompt tuning method (CoOp (Zhou et al., 2022)) in Appendix Sec. C.5. Further implementation details for all baselines are provided in Appendix Sec. A.

**Implementation Details.** Following prior works on adversarial robustness of VLMs (Zhou et al., 2024; Mao et al., 2023; Yu et al., 2024), we primarily use CLIP (Radford et al., 2021) with a ViT-B/32 image encoder (Dosovitskiy et al., 2020), tuning only binary mask parameters while keeping pre-trained weights frozen. We also report results using different encoder backbones (i.e., ViT-B/16, ViT-L/14) in Sec. C.3 and another VLM (i.e., VisualBERT (Li et al., 2019)) in Sec. C.4 of Appendix. Following Zheng et al. (2023), we apply learnable binary masks only to multi-head self-attention layers, comprising about 20% of vision encoder parameters, with other mask-tuning settings consistent with them. Our LAFA loss is applied across all transformer layers with coefficient $\lambda = 50.0$. Dataset-specific prompt templates (e.g., "a photo of a CLASS") are provided in Appendix (Sec. A). All results are averaged over three random seeds. For adversarial training, we use PGD (Madry et al., 2017) under the $l_\infty$ norm. During tuning, adversarial perturbations are generated with 2-step PGD ($\epsilon = \alpha = 1/255$); at test time, robustness is evaluated with 100-step PGD. Here, adversarial perturbations are computed by backpropagating through the masked model $f_{\theta \odot M_{bin}}$,

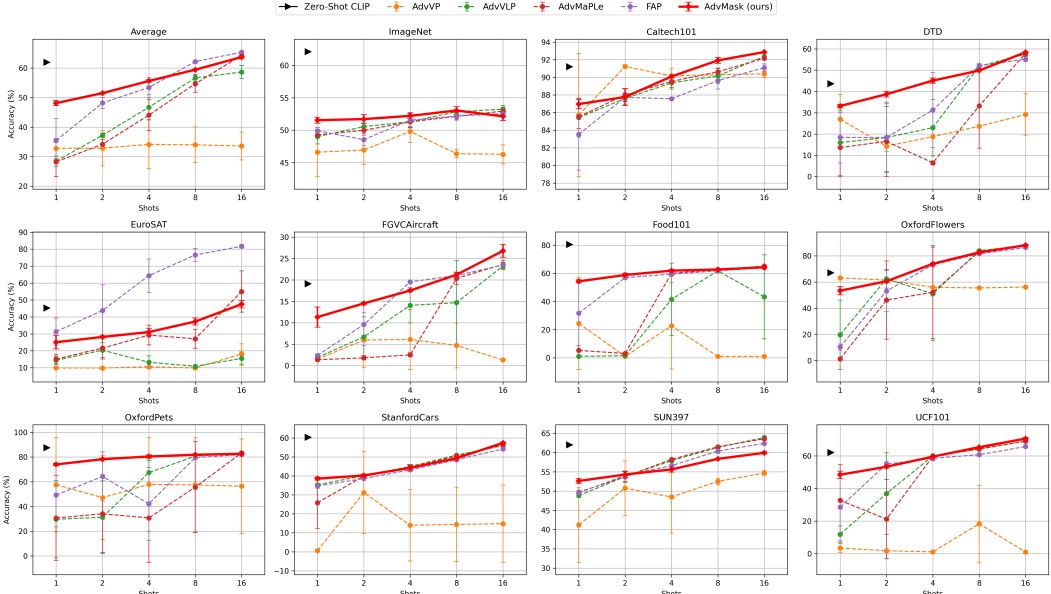

Figure 3: Clean test accuracy (%, ↑) over 11 datasets in few-shot settings. Results are averaged over 3 random trials (full results with standard deviations are provided in Sec. B.1 of Appendix).

ensuring a fully adaptive and fair evaluation setting. All baselines are implemented following their original settings. Additional hyperparameters and implementation details are given in Sec. A.

## 3.2 EXPERIMENTAL RESULTS

**Few-shot Adversarial Robustness.** In Fig. 2, we compare the adversarial robustness of our AdvMask method and baselines across 11 downstream datasets under the few-shot setting. As anticipated, the zero-shot CLIP model exhibits significant vulnerability to adversarial perturbations (average clean accuracy of approximately 61.9%, but adversarial accuracy drops drastically to about 2.5%). In contrast, our AdvMask consistently achieves substantially higher adversarial robustness than prompt-based and adapter-based baseline methods across most datasets, highlighting the effectiveness of our binary mask tuning strategy in selectively deactivating parameters vulnerable to adversarial attacks. Furthermore, AdvMask demonstrates robust performance across most few-shot settings, effectively enhancing adversarial robustness even with limited downstream data. This advantage stems from the fact that mask tuning efficiently explores task-specific pathways using few examples, while freezing pretrained weights preserves the strong generalizable representations of the base VLM. Additionally, our LAFA loss provides stable feature-level learning signals and mitigates noisy overfitting, making AdvMask particularly suitable for scarce-data scenarios. Consequently, our method is practical not only from the perspective of computational efficiency but also data efficiency, as it reliably tunes robust binary masks using only a small number of samples.

**Trade-off between Robustness and Transferability.** Another important requirement for adversarial tuning is maintaining the original clean accuracy on downstream tasks, as shown in Fig. 3. With extremely limited samples (e.g., 1, 2, or 4-shots), we observe an inevitable drop in clean accuracy, similar to other baseline methods, due to overfitting to the limited adversarial examples. However, as the number of tuning samples increases, clean accuracy gradually recovers, and surprisingly, even exceeds the original CLIP performance in certain cases (e.g., 8, 16-shots in Caltech101), despite the absence of explicit supervised loss on clean samples during mask tuning. We hypothesize that, given a moderate number of tuning samples (e.g., 16-shots), our method learns a generalizable binary mask that provides a regularizing effect, facilitating effective adaptation to downstream task without overfitting. Furthermore, AdvMask generally achieves higher clean accuracy than baseline methods, effectively balancing the trade-off between adversarial robustness and transferability. Therefore, our AdvMask offers practical benefits for reliable deployment of VLMs in real-world few-shot scenarios.

**Base-to-New Generalization Setting.** In Table 1, we present results under the base-to-new generalization setting, where classes are split into disjoint "base" (training) and "new" (testing) groups in each dataset. In the experiments, models are adversarially tuned with 16-shot samples from base classes and then evaluated on both groups. Even in this challenging scenario, AdvMask consistently achieves superior adversarial robustness while maintaining competitive clean accuracy. On base classes, it significantly improves adversarial accuracy over FAP with comparable clean performance. On unseen new classes, AdvMask outperforms all baselines in adversarial robustness and achieves the highest clean accuracy

| Method | Base Class | | New Class | | |
| | Clean | Adv. | Clean | Adv. | H |
|---|---|---|---|---|---|
| CLIP | 66.9 | 3.4 | **71.5** | 3.8 | 6.9 |
| AdvVP | 31.7 | 14.4 | 30.4 | 13.4 | 19.2 |
| AdvVLP | 59.0 | 32.4 | 46.9 | 21.6 | 34.6 |
| AdvMaPLe | 60.4 | 30.7 | 46.2 | 20.3 | 33.3 |
| FAP | **70.5** | 38.0 | 49.6 | 21.9 | 37.6 |
| **AdvMask** | 69.5 | **43.6** | 50.2 | **26.1** | **41.9** |

Table 1: Results on adversarial base-to-new generalization settings. For both class groups (base, new), we report the average clean and adversarial accuracy across 11 datasets, and the harmonic mean (H) of these four accuracy scores. (Detailed results are provided in Sec. B.2 of Appendix.)

(except zero-shot CLIP). These results confirm that AdvMask effectively identifies robust neural pathways by capturing inherent task-specific features from limited tuning samples, producing a robust and generalizable mask suitable for diverse test scenarios.

**Generalization Capability.** While our primary goal is to enhance robustness in few-shot adaptation scenarios, we emphasize that the learned mask also generalizes well to unseen datasets. To evaluate this, we follow the setup of TGA-ZSR (Yu et al., 2024), a recently proposed zero-shot robustness method. Specifically, we train AdvMask on a held-out source dataset (i.e., TinyImageNet) and directly testing on unseen target datasets without further tuning (Table 2). The results show that, in the 16-shot

| Method | Dataset | Clean Acc. (%) | Adv. Acc. (%) |
|---|---|---|---|
| CLIP | – | 61.9 (±0.0) | 2.7 (±0.0) |
| TGA-ZSR | Entire (100%) | 38.6 (±1.0) | 22.9 (±0.5) |
| FAP | 16-shot (3.2%) | 36.0 (±0.9) | 16.8 (±0.7) |
| TGA-ZSR | 16-shot (3.2%) | 41.3 (±1.0) | 13.0 (±0.3) |
| AdvMask | 16-shot (3.2%) | **42.0** (±0.3) | **19.4** (±0.2) |

Table 2: Results on zero-shot robustness. Following Yu et al. (2024), models are adapted using TinyImageNet (entire training set for TGA-ZSR, 16-shots for others) and evaluated on unseen downstream datasets. Results are averaged over 3 trials.

setting, AdvMask achieves superior clean and adversarial accuracy compared to baselines, demonstrating strong generalization capability in few-shot scenarios. Furthermore, despite using only 3.2% of the source data (i.e., 16 shots), AdvMask approaches the performance of TGA-ZSR, even though TGA-ZSR requires full access to the entire source dataset. These results suggest that AdvMask selectively deactivates parameters that are globally vulnerable to adversarial perturbations, rather than overfitting to dataset-specific patterns. In other words, certain parameters consistently amplify adversarial noise across tasks, and suppressing them yields more stable intermediate representations. This intuition is further supported by results from the base-to-new generalization setting with disjoint class groups. Consequently, AdvMask not only improves robustness on the tuned dataset but also produces transferable masks effective for unseen domains, making it well-suited for robust and reliable deployment in real-world applications.

### 3.3 FURTHER STUDIES ON ADVMASK

In this section, we provide ablation studies and additional analyses of AdvMask. We report 16-shot results averaged over 3 random trials on 5 datasets (i.e., Caltech101, DTD, FGVCAircraft, Flowers102, UCF101) from diverse categories. Due to page limits, extended results are provided in the Appendix (Sec. C & Sec. D), including ablations on key design choices (e.g., *LAFA loss coefficient, layer positions, adaptive weighting scheme, mask threshold $\alpha$*), robustness under *different perturbation bounds and attack type*, evaluations on *alternative architectures and VLMs*, as well as complementary analyses on *compatibility with learnable prompt tuning* and *computational efficiency*.

**Which Layers are Effective for Adversarial Masking?** Since tuning masks for all parameters within CLIP incurs substantial computational costs, we specifically focus on optimizing masks for the multi-head attention (MHSA) layers within the transformer blocks of the image encoder, following prior work (Zheng et al., 2023). Our choice is motivated by the well-established observation that these self-attention layers generate context-aware representations by capturing long-range dependencies across input tokens (i.e., image patches), making them particularly vulnerable to ad-

versarial perturbations in the input space. Therefore, selectively masking noise-sensitive parameters within these self-attention layers proves highly effective as shown in Table 3. Also, as these layers comprise only about 20% of the total parameters (significantly fewer than the MLP layers), our approach significantly reduces computational costs, including memory usage and training time. Consequently, our AdvMask offers a practical and effective strategy for parameter-efficient tuning of large-scale VLMs, making it highly suitable for diverse real-world applications.

| Module | Clean Acc. (%) | Adv. Acc. (%) |
|---|---|---|
| MLP only | $65.73 \pm 0.45$ | $45.95 \pm 0.07$ |
| MHSA only | $\mathbf{67.34 \pm 0.19}$ | $47.13 \pm 0.25$ |
| MHSA + MLP | $66.01 \pm 0.28$ | $\mathbf{47.20 \pm 0.25}$ |

Table 3: Ablation on adversarial masking layers.

**Loss Ablation Study.** In Table 4, we present ablations on the loss functions. Our design combines adversarial cross-entropy ($\mathcal{L}_{\text{CE-adv}}$) with the LAFA loss to enforce feature-level consistency between clean and adversarial samples. To validate this choice, we compare LAFA with several alternative auxiliary losses (e.g., Jensen-Shannon divergence, KL divergence). The results show that $\mathcal{L}_{\text{LAFA}}$ consistently outperforms these alternatives in both clean accuracy and adversarial robustness, with the performance gap becoming more pronounced in low-shot settings (e.g., 1-shot and 4-shot). This is attributed to the fact that LAFA's feature-alignment objective provides a stronger and more stable learning signal than distributional divergence terms defined in the output space, allowing the model to maintain coherent and robust representations even when data is scarce.

Moreover, the adaptive weighting scheme (Sec. 2.3) further improves performance by emphasizing reliable samples during training. This mechanism mitigates overfitting to noisy or misclassified examples, and its benefits are particularly apparent in low-shot scenarios (see Appendix Sec. D.2 for details). Overall, these findings confirm that our loss formulation enhances representational robustness and enables AdvMask to reliably identify robust neural pathways, even under challenging fewshot conditions.

| Loss Function | 1-shot Clean | 1-shot Adv | 4-shot Clean | 4-shot Adv | 16-shot Clean | 16-shot Adv |
|---|---|---|---|---|---|---|
| CLIP | 56.6 | 4.8 | 56.6 | 4.8 | 56.6 | 4.8 |
| $\mathcal{L}_{\text{CE-adv}}$ | 40.3 | 15.6 | 55.2 | 30.6 | 65.8 | 46.4 |
| $+ \mathcal{L}_{\text{JS}}$ | 42.9 | 17.3 | 54.7 | 30.8 | 65.9 | 46.5 |
| $+ \mathcal{L}_{\text{KL}}$ | 31.8 | 13.8 | 46.2 | 27.9 | 60.7 | 43.6 |
| $+ \mathcal{L}_{\text{LAFA}}^{\dagger}$ | $\mathbf{44.5}$ ($\pm 1.51$) | $\mathbf{17.8}$ ($\pm 0.97$) | $\mathbf{56.6}$ ($\pm 0.68$) | $\mathbf{32.1}$ ($\pm 0.64$) | $\mathbf{66.9}$ ($\pm 0.44$) | $\mathbf{46.8}$ ($\pm 0.41$) |
| $+ \mathcal{L}_{\text{LAFA}}$ | $\mathbf{46.6}$ ($\pm 1.11$) | $\mathbf{18.4}$ ($\pm 0.45$) | $\mathbf{57.2}$ ($\pm 0.34$) | $\mathbf{32.2}$ ($\pm 0.28$) | $\mathbf{67.3}$ ($\pm 0.19$) | $\mathbf{47.1}$ ($\pm 0.25$) |

Table 4: Loss ablation with alternative auxiliary losses. $\mathcal{L}_{\text{LAFA}}^{\dagger}$ represents LAFA loss without adaptive weighting scheme.

**Further Insight into Robust Neural Pathway.** A neural pathway refers to a propagation path within a pre-trained network that forms new functional conjunctions between neurons (or learned knowledge) (Zheng et al., 2023). Under the lottery ticket hypothesis, Malach et al. (2020) showed that mask optimization within overparameterized networks can achieve performance comparable to full weight optimization. Motivated by these works, we explore whether mask-based optimization can uncover pathways that remain robust under adversarial attack. Intuitively, small adversarial perturbations, though imperceptible in the input space, can amplify through deeper layers of an encoder, leading to

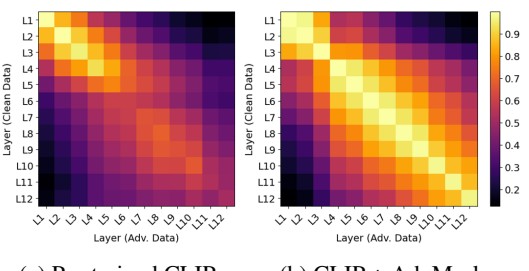

(a) Pre-trained CLIP     (b) CLIP + AdvMask

Figure 4: Layer-wise CKA similarity between clean and adversarial features on *DTD* dataset, propagated from L1 to L12 (i.e., top-left to bottom-right).

incorrect predictions. Our binary masks mitigate this by deactivating vulnerable weights, suppressing noise propagation and preserving stable intermediate representations. To support this, we analyze layer-wise representations using CKA similarity (Fig. 4), a well-known metric for representation consistency (Kornblith et al., 2019). We measure how similar clean and adversarial features (CLS tokens) remain across layers, with and without AdvMask. It shows that pre-trained CLIP shows high similarity in early layers (L1-L4) but declines in deeper ones as adversarial noise amplifies. In contrast, AdvMask preserves consistently higher similarity across all layers, effectively suppressing noise amplification and stabilizing representations. This provides strong evidence that AdvMask identifies robust neural pathways that enhance adversarial resilience.

## 4 RELATED WORKS

**Parameter-Efficient Adaptation Methods for VLMs.** Vision-language models (VLMs) such as CLIP (Radford et al., 2021) show strong transferability across diverse tasks (Zhang et al., 2024b), but their scale makes full fine-tuning impractical. This has motivated parameter-efficient approaches, including text, visual, and joint prompt tuning (Zhou et al., 2022; Bahng et al., 2022; Khattak et al., 2023), as well as adapter methods (Zhang et al., 2022b; Gao et al., 2024). Recently, mask tuning (Zheng et al., 2023) has been proposed to identify task-specific subnetworks within pre-trained VLMs. However, they mainly target downstream accuracy, leaving adversarial robustness unexplored. In contrast, we develop mask tuning explicitly for robustness by uncovering robust neural pathways.

**Adversarial Robustness for VLMs.** Despite their generalization, VLMs are highly vulnerable to adversarial attacks (Cui et al., 2024; Budathoki & Dhakal, 2025), limiting real-world deployment. Prior efforts include adversarial prompt tuning (Zhou et al., 2024; Mao et al., 2023), which improves robustness but ignores the encoder's intrinsic structure. Fully fine-tuning with adversarial training (Bai et al., 2021) is effective but costly and prone to overfitting in few-shot settings. Zero-shot robustness methods (Yu et al., 2024) rely on held-out datasets but often fail under distribution shifts. We instead focus on parameter-efficient adversarial tuning in few-shot scenarios, achieving robustness gains with minimal data while preserving pre-trained weights.

**Neural Pathways Searching.** Deep networks distribute knowledge across neurons, dynamically forming task-specific pathways (Liu et al., 2018; Zhao et al., 2020). Building on this perspective, binary mask tuning has been explored as a means to isolate subnetworks for task adaptation (Wortsman et al., 2020; Csordás et al., 2020) or for addressing OOD generalization (Zhang et al., 2021). Recent work (Zheng et al., 2023) further demonstrated that mask tuning can reveal latent knowledge within pretrained VLMs, though robustness under adversarial perturbations remains unaddressed. Our work extends this line of research by introducing adversarial mask tuning to deactivate noise-sensitive parameters, thereby constructing robust neural pathways and substantially improving VLM robustness. Related efforts in adversarial learning have explored robustness-sensitive structures from different angles. Adversarial pruning (AP) methods (Piras et al., 2025; Sehwag et al., 2020; Chen et al., 2022) aim to obtain sparse yet robust models by pruning and retraining weights, which contrasts with our goal of robust few-shot adaptation of pretrained VLMs without modifying any weights. Also, Zhu et al. (2023) similarly analyzes robustness-critical components but focuses on improving the generalization of adversarially trained models through fine-tuning, whereas AdvMask learns binary masks on pretrained VLMs (without adversarial pretraining or weight updates) to achieve robust adaptation in few-shot scenarios.

## 5 CONCLUSION

In this paper, we introduced AdvMask, a framework that uncovers robust neural pathways in VLMs for few-shot adaptation. By introducing the LAFA loss to adaptively align clean and adversarial features, AdvMask selectively emphasizes robust representations through binary masks, enabling efficient and reliable adaptation without altering pre-trained weights. Extensive experiments confirm AdvMask's effectiveness over prior adversarial tuning methods, offering a new perspective that robust subnetworks inherently exist within large VLMs. These findings highlight a path toward more resilient and parameter-efficient deployment of robust models in real-world applications.

## REPRODUCIBILITY

To ensure reproducibility, we provide our implementation code in `https://github.com/wonjeongchoi/AdvMask` and the supplementary materials. Further experimental settings and detailed configurations, including computing resources, are described in Sec. 3.1 of the main paper and Sec. A.3 of Appendix.

ACKNOWLEDGEMENTS

This work was supported by the National Research Foundation of Korea (NRF) through grants from the Korea government (MSIT) (No. RS-2024-00408003, RS-2024-00340966) and by the Institute of Information & communications Technology Planning & Evaluation (IITP) grant funded by MSIT of Korea (No. RS-2024-00444862). Also, it was partly supported by the Technology development Program of MSS (No. RS-2025-25464351). Dong-Jun Han is the corresponding author.

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

APPENDIX

# A EXPERIMENTAL DETAILS

## A.1 DATASETS

Our few-shot experiments are conducted on 11 public datasets for image classification tasks, following Zhou et al. (2022; 2024). For experiments on zero-shot adversarial robustness, we adopt TinyImageNet (Le & Yang, 2015) as the source dataset for tuning and subsequently evaluate the tuned model on the other downstream datasets, following Yu et al. (2024). To facilitate understanding, in Table 5, we summarize the statistics of datasets used in our experiments. Additionally, for baselines requiring static textual prompts (e.g., "a photo of a {CLASS}") for adversarial tuning such as zero-shot CLIP, AdvVP, and our AdvMask, we specify hand-crafted text prompt templates for each dataset in Table 6. We note that these templates are also used as initial prompts in learnable text prompt tuning methods, including AdvVLP, AdvMaPLe, and FAP.

| Dataset | #Classes | Train Size | Test Size | Task |
|---|---|---|---|---|
| ImageNet | 1,000 | 1.28M | 50,000 | Object recognition |
| TinyImageNet | 200 | 0.1M | 10,000 | Object recognition |
| Caltech101 | 100 | 4,128 | 2,465 | Object recognition |
| DTD | 47 | 2,820 | 1,692 | Texture recognition |
| EuroSAT | 10 | 13,500 | 8,100 | Satellite image recognition |
| FGVCAircraft | 100 | 3,334 | 3,333 | Fine-grained aircraft recognition |
| Flowers102 | 102 | 4,093 | 2,463 | Fine-grained flowers recognition |
| Food101 | 101 | 50,500 | 30,300 | Fine-grained food recognition |
| OxfordPets | 37 | 2,944 | 3,669 | Fine-grained pets recognition |
| StanfordCars | 196 | 6,509 | 8,041 | Fine-grained car recognition |
| SUN397 | 397 | 15,880 | 19,850 | Scene recognition |
| UCF101 | 101 | 7,639 | 3,783 | Action recognition |

Table 5: Summary of datasets, including number of classes, training/testing sizes, and task types.

| Dataset | Text Template |
|---|---|
| ImageNet | `"a photo of a {CLASS}."` |
| TinyImageNet | `"a photo of a {CLASS}."` |
| Caltech101 | `"a photo of a {CLASS}."` |
| DTD | `"{CLASS} texture."` |
| EuroSAT | `"a centered satellite photo of {CLASS}."` |
| OxfordPets | `"a photo of a {CLASS}, a type of pet."` |
| FGVCAircraft | `"a photo of a {CLASS}, a type of aircraft."` |
| Food101 | `"a photo of a {CLASS}, a type of food."` |
| Flowers102 | `"a photo of a {CLASS}, a type of flower."` |
| StanfordCars | `"a photo of a {CLASS}."` |
| SUN397 | `"a photo of a {CLASS}."` |
| UCF101 | `"a photo of a person doing {CLASS}."` |

Table 6: Hand-crafted text templates across different datasets.

## A.2 IMPLEMENTATION DETAILS FOR BASELINE METHODS

**Adversarial Prompt Tuning.** Similar to (Zhou et al., 2024), we implement adversarial prompt-based baselines, strictly following the original architectural and parameter settings for fair comparison. Specifically, adversarial visual prompts (AdvVP) adopt a token-level prompt of size 5 and a 30-pixel padding around the image, optimized for 10 epochs using SGD with a cosine learning rate scheduler (initial learning rate: 40), following the setup of (Mao et al., 2023). Adversarial multi-modal prompts (AdvMaPLe) employ token-level prompts of size 2 in both text and visual branches for the first 9 transformer layers, coupled with text-to-image projections. Adversarial vision-language prompts

(AdvVLP) use an identical structure but adapt vision and language prompts independently. Both AdvMaPLe and AdvVLP are trained for 10 epochs with SGD and a cosine scheduler (initial learning rate: 0.0035). For consistency, we replace the original baseline loss functions with the adversarial TeCoA (Mao et al., 2023) loss during training and evaluation for AdvVP, AdvMaPLe, and AdvVLP. For the state-of-the-art method of few-shot adversarial prompt tuning (FAP) (Zhou et al., 2024), we train the model for 10 epochs by using SGD with a momentum of 0.9 and a cosine scheduler (initial learning rate: 0.0035) with a warm-up strategy during the first epoch. Also, we use token prompts of size 2 in both branches for the first 9 layers, following the configurations in original paper.

### A.3 Additional Implementation Details

For our AdvMask, all elements of mask parameters are initialized with $10^{-2}$ and the threshold $\alpha$ (in Eqn. (2)) is set to $5 \times 10^{-3}$, following Zheng et al. (2023). Regarding the optimization setup, we train the (binary) mask parameters using a SGD optimizer with a momentum of 0.9 and a cosine scheduler with a warm-up strategy during the first epoch, following the setup of (Zhou et al., 2024). For most of the datasets, models are trained for 10 epochs with the initial learning rate of 0.01. For ImageNet, Food101, and SUN397 datasets, considering large number of classes and data volumes, we use the learning rate of 0.0035 and maximum epochs of 10, except for ImageNet with 5 epochs. In the experiments on zero-shot adversarial robustness, all models are trained for 5 epochs with the same configurations with a few-shot settings. We conduct all experiments in an environment with PyTorch 1.12.1 and CUDA 11.3 on Python 3.8 under a single NVIDIA RTX 3090 GPU (24GB) device.

## B Comprehensive Results

### B.1 Results under Few-Shot Settings

| Shots | ImageNet | Caltech | DTD | EuroSAT | FGVC | Food101 | Flowers | Pets | Cars | SUN397 | UCF101 | Avg. |
|---|---|---|---|---|---|---|---|---|---|---|---|---|
| 1 | 51.53 ($\pm$0.46) | 86.97 ($\pm$0.54) | 33.17 ($\pm$0.74) | 25.07 ($\pm$4.03) | 11.33 ($\pm$2.35) | 54.30 ($\pm$1.13) | 53.37 ($\pm$3.15) | 73.97 ($\pm$0.95) | 38.57 ($\pm$0.82) | 52.63 ($\pm$0.61) | 48.40 ($\pm$2.24) | 48.12 ($\pm$0.86) |
| 2 | 51.70 ($\pm$0.70) | 87.77 ($\pm$0.97) | 38.70 ($\pm$1.36) | 28.17 ($\pm$0.71) | 14.53 ($\pm$0.24) | 58.83 ($\pm$1.27) | 60.57 ($\pm$0.90) | 78.23 ($\pm$0.84) | 40.23 ($\pm$0.85) | 54.30 ($\pm$0.79) | 53.30 ($\pm$0.67) | 51.48 ($\pm$0.24) |
| 4 | 52.20 ($\pm$0.43) | 90.10 ($\pm$0.16) | 45.10 ($\pm$1.24) | 31.07 ($\pm$2.21) | 17.53 ($\pm$0.26) | 61.90 ($\pm$0.80) | 73.87 ($\pm$0.59) | 80.40 ($\pm$1.40) | 44.23 ($\pm$1.73) | 55.63 ($\pm$0.74) | 59.50 ($\pm$0.43) | 55.59 ($\pm$0.42) |
| 8 | 53.03 ($\pm$0.61) | 91.93 ($\pm$0.34) | 49.83 ($\pm$0.09) | 37.27 ($\pm$2.09) | 21.23 ($\pm$0.59) | 62.73 ($\pm$0.33) | 82.60 ($\pm$1.31) | 81.83 ($\pm$0.73) | 49.33 ($\pm$1.51) | 58.37 ($\pm$0.05) | 65.33 ($\pm$0.12) | 59.41 ($\pm$0.17) |
| 16 | 52.13 ($\pm$0.65) | 92.87 ($\pm$0.05) | 58.43 ($\pm$0.17) | 47.53 ($\pm$2.22) | 26.80 ($\pm$1.50) | 64.27 ($\pm$0.46) | 88.03 ($\pm$0.61) | 82.63 ($\pm$0.21) | 57.40 ($\pm$0.42) | 59.93 ($\pm$0.29) | 70.57 ($\pm$0.86) | 63.69 ($\pm$0.23) |

Table 7: Clean test accuracy (%, ↑) of our AdvMask over 11 datasets in few-shot settings. Results are averaged over 3 random trials.

In Table 7 and Table 8, we present the complete few-shot results of our AdvMask on clean and adversarial samples across 11 datasets, respectively. These results align with the main findings in Fig. 2 and Fig. 3 of our main paper, demonstrating that AdvMask achieves superior adversarial robustness compared to baseline methods while effectively balancing the trade-off between robustness and transferability. Additionally, our method exhibits low standard deviations (on average lower than 1.0%) across datasets, highlighting the stability and effectiveness of AdvMask in identifying robust neural pathways, even under challenging few-shot scenarios.

### B.2 Results under Base-to-New Generalization Settings

In Table 9, we provide the complete results of our AdvMask method in base-to-new generalization settings across 11 datasets, consistent with Table 1 in the main paper. Even in this challenging scenario, where the generalization capability of the adapted model is important, our AdvMask still achieves competitive performance by effectively capturing inherent task-specific features from limited samples. This demonstrates our AdvMask's strong generalization capability for large-scale test datasets.

| Shots | ImageNet | Caltech | DTD | EuroSAT | FGVC | Food101 | Flowers | Pets | Cars | SUN397 | UCF101 | Avg. |
|---|---|---|---|---|---|---|---|---|---|---|---|---|
| 1 | 14.27 (±0.33) | 47.60 (±0.86) | 9.93 (±0.50) | 5.57 (±4.76) | 1.93 (±0.78) | 7.03 (±0.74) | 20.87 (±0.25) | 9.40 (±0.85) | 6.40 (±0.16) | 15.80 (±0.24) | 11.53 (±2.45) | 13.67 (±0.51) |
| 2 | 16.95 (±0.25) | 56.93 (±1.72) | 15.93 (±1.08) | 12.90 (±2.18) | 3.53 (±0.29) | 12.90 (±0.29) | 27.17 (±0.62) | 18.43 (±1.23) | 8.70 (±0.16) | 19.80 (±0.24) | 19.50 (±1.36) | 19.34 (±0.25) |
| 4 | 19.30 (±0.24) | 65.30 (±0.70) | 21.63 (±1.11) | 20.50 (±1.84) | 5.57 (±0.34) | 17.50 (±0.22) | 41.43 (±0.69) | 28.37 (±0.74) | 12.23 (±0.31) | 23.80 (±0.36) | 27.03 (±0.34) | 25.70 (±0.31) |
| 8 | 20.90 (±0.50) | 71.13 (±0.73) | 27.70 (±0.57) | 27.90 (±0.54) | 7.47 (±0.12) | 23.40 (±0.14) | 57.37 (±0.26) | 38.23 (±0.26) | 18.23 (±0.82) | 28.23 (±0.25) | 34.13 (±0.53) | 32.24 (±0.14) |
| 16 | 23.77 (±0.56) | 75.83 (±0.21) | 35.47 (±1.23) | 32.73 (±0.74) | 12.03 (±0.49) | 28.00 (±0.62) | 69.90 (±0.29) | 44.73 (±0.21) | 27.37 (±0.12) | 31.00 (±0.14) | 42.43 (±0.29) | 38.48 (±0.22) |

Table 8: Adversarial test accuracy (%, ↑) of our AdvMask over 11 datasets in few-shot settings. Results are averaged over 3 random trials.

| Class | Type | ImageNet | Caltech | DTD | EuroSAT | FGVC | Food101 | Flowers | Pets | Cars | SUN397 | UCF101 | Avg. |
|---|---|---|---|---|---|---|---|---|---|---|---|---|---|
| Base | Clean | 56.53 (±0.37) | 95.73 (±0.50) | 70.07 (±1.33) | 66.63 (±2.34) | 25.23 (±0.87) | 69.97 (±0.21) | 91.10 (±0.37) | 87.27 (±0.39) | 57.77 (±0.42) | 68.43 (±0.45) | 75.47 (±1.45) | 69.47 (±0.79) |
| | Adv. | 26.70 (±0.57) | 81.07 (±0.47) | 41.40 (±1.59) | 55.40 (±2.34) | 11.07 (±0.52) | 30.60 (±0.22) | 75.33 (±0.46) | 49.53 (±0.84) | 25.43 (±0.53) | 36.80 (±0.29) | 46.27 (±0.29) | 43.60 (±0.74) |
| New | Clean | 47.80 (±0.80) | 84.47 (±1.07) | 45.27 (±0.09) | 31.17 (±3.27) | 13.20 (±0.08) | 63.10 (±2.20) | 41.53 (±1.07) | 84.03 (±0.87) | 34.70 (±0.08) | 59.20 (±0.78) | 47.23 (±0.68) | 50.15 (±1.00) |
| | Adv. | 21.50 (±0.08) | 61.33 (±0.45) | 22.73 (±1.37) | 22.60 (±2.27) | 4.27 (±0.48) | 25.53 (±0.73) | 19.47 (±0.83) | 47.90 (±1.70) | 11.33 (±0.34) | 29.57 (±0.66) | 20.93 (±0.62) | 26.11 (±0.87) |

Table 9: Results on adversarial base-to-new generalization settings. For both class groups (base, new), we report the clean and adversarial accuracy (mean ± standard deviation) across 11 datasets. Models are tuned using 16-shot samples from the base class group.

## B.3 RESULTS UNDER ZERO-SHOT ROBUSTNESS SETTINGS

In Table 10 and Table 11, we provide zero-shot results on clean and adversarial samples across downstream datasets. As described in Table 2 of our main paper, we first adversarially tune the model using TinyImageNet as the source dataset and subsequently evaluate the tuned model on 10 downstream datasets. For TGA-ZSR (Yu et al., 2024), a state-of-the-art zero-shot adversarial robustness method, we use the entire source training set (100%), while other methods utilize only 16-shot samples (3.2%) for tuning. Notably, even with significantly fewer samples, our AdvMask achieves competitive zero-shot performance on both clean and adversarial samples. Specifically, for the source dataset (i.e., TinyImageNet), our accuracy scores are inevitably lower than TGA-ZSR due to fewer training samples. However, on downstream datasets, our AdvMask attains better clean accuracy and only slightly lower adversarial accuracy (approximately 2.9% lower on average) compared to TGA-ZSR, despite using only 3.2% of the source data. Furthermore, AdvMask significantly outperforms FAP (Zhou et al., 2024) in downstream tasks, highlighting its superior zero-shot generalization from limited tuning samples. Additionally, unlike TGA-ZSR's resource-intensive full-parameter adaptation, AdvMask optimizes only binary masks, considerably enhancing efficiency in terms of memory usage and training latency. Therefore, our method is practical and effective for both few-shot and zero-shot scenarios, enabling robust and reliable deployment of VLMs in real-world applications.

## C ADDITIONAL RESULTS

### C.1 ROBUSTNESS UNDER DIFFERENT PERTURBATION BOUNDS

For comprehensive evaluation, in Fig. 5, we provide additional results under varying perturbation bounds (i.e., $\epsilon$ in Eqn. (5) of the main paper). We compare AdvMask with other promising methods (i.e., zero-shot CLIP, FAP) in the few-shot scenario using 16-shot samples. From the results, we observe that even as stronger adversarial attacks occur with increased perturbation bounds, our AdvMask consistently achieves competitive adversarial robustness. Although FAP achieves

| Method | Dataset | Source T-ImgNet | Downstream Datasets Caltech101 | DTD | EuroSAT | FGVC | Food101 | Flowers | Pets | Cars | SUN397 | UCF101 | Avg. |
|---|---|---|---|---|---|---|---|---|---|---|---|---|---|
| CLIP | – | 61.20 | 91.20 | 43.60 | 45.20 | 19.10 | 80.50 | 67.00 | 87.50 | 60.40 | 62.00 | 62.00 | 61.85 |
| TGA-ZSR | Entire (100%) | 79.83 (±0.74) | 85.60 (±0.93) | 23.87 (±1.64) | 17.23 (±1.25) | 6.27 (±0.42) | 40.07 (±0.78) | 33.03 (±0.61) | 61.63 (±1.92) | 26.30 (±1.71) | 44.20 (±1.67) | 48.13 (±1.16) | 38.63 (±1.00) |
| FAP | 16-shot (3.2%) | 53.37 (±0.29) | 81.60 (±0.78) | 19.63 (±2.50) | 18.63 (±1.22) | 5.63 (±0.59) | 34.70 (±1.34) | 33.37 (±1.92) | 66.90 (±1.71) | 22.30 (±3.48) | 37.90 (±2.12) | 39.23 (±1.84) | 35.99 (±0.89) |
| TGA-ZSR | 16-shot (3.2%) | 67.03 (±0.41) | 81.53 (±1.85) | 24.23 (±1.92) | 21.03 (±0.87) | 8.23 (±0.48) | 50.37 (±1.37) | 32.90 (±1.85) | 60.20 (±2.49) | 34.70 (±1.40) | 47.10 (±0.99) | 52.50 (±0.50) | 41.28 (±1.04) |
| AdvMask | 16-shot (3.2%) | 59.07 (±0.37) | 84.47 (±0.21) | 28.27 (±0.78) | 21.90 (±2.41) | 9.13 (±0.25) | 41.33 (±1.43) | 40.47 (±0.33) | 69.80 (±1.15) | 33.53 (±1.03) | 45.03 (±0.31) | 45.97 (±0.45) | **41.99** (±0.34) |

Table 10: Results on zero-shot *clean accuracy*. All models are tuned using TinyImageNet as the source dataset (TGA-ZSR (Yu et al., 2024) uses the full training set, whereas other methods use 16-shot samples (3.2%) from the source dataset, not from downstream datasets). After tuning, models are evaluated zero-shot on 10 unseen downstream datasets. The average accuracy in the last column is computed over the 10 datasets across 3 trials.

| Method | Dataset | Source T-ImgNet | Downstream Datasets Caltech101 | DTD | EuroSAT | FGVC | Food101 | Flowers | Pets | Cars | SUN397 | UCF101 | Avg. |
|---|---|---|---|---|---|---|---|---|---|---|---|---|---|
| CLIP | – | 0.20 | 16.63 | 4.93 | 0.03 | 0.00 | 0.50 | 1.43 | 0.97 | 0.10 | 1.00 | 1.00 | 2.66 |
| TGA-ZSR | Entire (100%) | 52.87 (±0.58) | 67.73 (±0.76) | 15.70 (±1.31) | 11.33 (±0.12) | 3.10 (±0.45) | 17.03 (±0.54) | 18.43 (±0.78) | 36.00 (±0.37) | 12.23 (±0.65) | 20.77 (±0.53) | 26.63 (±0.66) | 22.90 (±0.51) |
| FAP | 16-shot (3.2%) | 18.63 (±0.69) | 55.77 (±1.18) | 11.53 (±1.56) | 10.30 (±1.02) | 1.70 (±0.36) | 9.80 (±0.16) | 15.27 (±1.97) | 30.50 (±1.56) | 5.77 (±0.60) | 12.20 (±0.91) | 15.10 (±0.71) | 16.79 (±0.67) |
| TGA-ZSR | 16-shot (3.2%) | 15.90 (±0.43) | 47.87 (±0.98) | 9.07 (±0.46) | 7.73 (±0.97) | 1.83 (±0.39) | 7.83 (±0.05) | 11.13 (±1.01) | 17.37 (±1.16) | 5.10 (±0.22) | 8.77 (±0.12) | 13.10 (±0.43) | 12.98 (±0.33) |
| AdvMask | 16-shot (3.2%) | 26.23 (±0.29) | 61.27 (±0.53) | 16.10 (±0.57) | 5.70 (±2.60) | 1.87 (±0.17) | 12.93 (±0.93) | 19.43 (±0.45) | 32.37 (±0.66) | 8.17 (±0.12) | 16.40 (±0.28) | 19.33 (±0.87) | **19.36** (±0.25) |

Table 11: Results on zero-shot *adversarial accuracy*. All models are tuned using TinyImageNet as the source dataset (TGA-ZSR (Yu et al., 2024) uses the full training set, whereas other methods use 16-shot samples (3.2%) from the source dataset, not from downstream datasets). After tuning, models are evaluated zero-shot on 10 unseen downstream datasets. The average accuracy in the last column is computed over the 10 datasets across 3 trials.

the highest clean accuracy due to their explicit supervision on clean samples during training, its adversarial robustness notably deteriorates under larger perturbations, whereas AdvMask remains robust against stronger attacks. These results confirm that AdvMask's robustness gains persist even under stronger attacks (e.g., $\epsilon = 4/255$), indicating that the binary mask and straight-through estimator do not obscure gradients. Therefore, our approach represents an effective and practical solution for deployment in reliable systems where resistance to dynamic adversarial attacks is crucial.

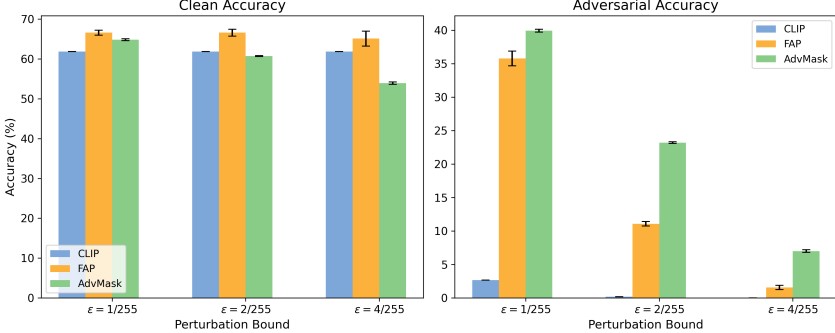

Figure 5: Results under varying perturbation bounds (i.e., $\epsilon$) in the few-shot scenario using 16-shot samples. We report the average clean and adversarial accuracy across 10 datasets over 3 trials.

## C.2 ROBUSTNESS UNDER DIFFERENT ATTACK TYPE

In Fig. 6, we conduct additional experiments to evaluate the adversarial robustness of our AdvMask under different attack type. Specifically, we apply AutoAttack (Croce & Hein, 2020), a stronger and user-independent attack strategy designed to overcome limitations (e.g., sub-optimal step sizes) of previous PGD-based attacks. Following Zhou et al. (2024), we consider two variants of APGD (i.e., APGD-CE and APGD-DLR) and compare our AdvMask with FAP, since other methods exhibit near-zero accuracy due to the stronger attack. Experiments are performed in a 16-shot scenario with a perturbation bound of $\epsilon = 1/255$ across 5 datasets. The results demonstrate that while FAP improves adversarial robustness over the zero-shot CLIP model, our AdvMask consistently outperforms this baseline, confirming the effectiveness of our robust mask-tuning approach under stronger attacks.

Additionally, we evaluate robustness against text-level and joint image–text-level adversarial attacks in Table 12. Specifically, by using the masks trained with 16-shot downstream samples, we assess whether the learned masks (although trained only with image-level adversarial supervision) can still provide robustness when different modalities are attacked. We consider two additional evaluation settings beyond standard image-level PGD attacks: (1) Independent multimodal attacks, where PGD is applied to the image while BERT-Attack Li et al. (2020) perturbs the text prompts; (2) Joint multimodal attacks, following CoAttack Zhang et al. (2022a), where image and text embeddings are perturbed in a coordinated manner within the shared multimodal space. As shown in Table 12, performance decreases under stronger multimodal attack scenarios, but our AdvMask still maintains robustness even though it was never trained with text or joint-level perturbations. These results suggest that our mask-based robustness transfer generalizes beyond image-level perturbations. Consequently, we conclude that AdvMask reliably identifies inherently robust neural pathways within VLMs, ensuring resilience against diverse adversarial attack types.

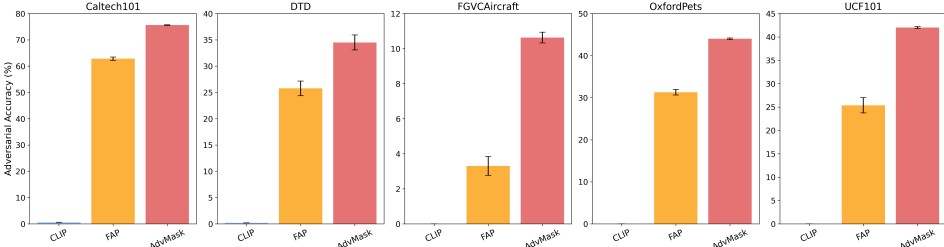

Figure 6: Adversarial robustness under AutoAttack. We conduct experiments in the 16-shot scenario with a perturbation bound of $\epsilon = 1/255$. We report the adversarial accuracy across 5 datasets, averaged over 3 trials.

| Attack Type | Accuracy (%) |
|---|---|
| CLIP (zero-shot baseline) | 7.30 |
| PGD + BERT-Attack (independent) | 29.20 |
| PGD + BERT-Attack (CoAttack-style joint attack) | 28.50 |

Table 12: Robustness of AdvMask under multimodal adversarial attacks. By using the masks trained with 16-shot downstream samples, we assess whether the learned masks can still provide robustness when different modalities are attacked. We report average adversarial accuracy (%, ↑) over 5 datasets across three different runs.

## C.3 ROBUSTNESS UNDER DIFFERENT BACKBONE ARCHITECTURES

In Table 13 and Table 14, we provide results using larger CLIP image encoders (e.g., ViT-B/16, ViT-L/14). Our AdvMask still yields significant gains in adversarial robustness over the most competitive baseline (i.e., FAP), demonstrating strong generalizability. We also note that unlike many prompt-based methods that rely on architecture-specific components (e.g., context tokens), our AdvMask is applicable to any vision encoder as long as intermediate features can be extracted for LAFA loss. This makes it broadly applicable for diverse downstream tasks and real-world scenarios.

Table 13: Results on CLIP ViT-B/16 encoder. Using ViT-B/16 as CLIP image encoder, we report 16-shot test accuracy (%, ↑) averaged over 5 datasets with 3 random trials.

| Method | Clean Accuracy (%) | | | | | | Adversarial Accuracy (%) | | | | | |
|---|---|---|---|---|---|---|---|---|---|---|---|---|
| | Caltech. | DTD | FGVC | Flowers | UCF. | **Avg.** | Caltech. | DTD | FGVC | Flowers | UCF. | **Avg.** |
| CLIP (ViT-B/16) | 92.9 | 44.4 | 24.8 | 71.4 | 66.7 | 60.0 | 5.8 | 1.6 | 0.0 | 0.1 | 0.1 | 1.5 |
| FAP | 92.3 | 60.6 | 26.6 | 84.7 | 69.9 | 66.8 | 61.0 | 26.8 | 6.1 | 49.4 | 26.3 | 33.9 |
| **AdvMask (ours)** | 90.7 | 63.3 | 31.4 | 90.2 | 68.8 | **68.9** | 77.2 | 37.3 | 14.4 | 76.5 | 45.9 | **50.3** |

Table 14: Results on CLIP ViT-L/14 encoder. Using ViT-L/14 as CLIP image encoder, we report 16-shot test accuracy (%, ↑) averaged over 5 datasets with 3 random trials.

| Method | Clean Accuracy (%) | | | | | | Adversarial Accuracy (%) | | | | | |
|---|---|---|---|---|---|---|---|---|---|---|---|---|
| | Caltech. | DTD | FGVC | Flowers | UCF. | **Avg.** | Caltech. | DTD | FGVC | Flowers | UCF. | **Avg.** |
| CLIP (ViT-L/14) | 95.2 | 53.0 | 32.5 | 79.2 | 75.0 | 67.0 | 13.7 | 3.0 | 0.0 | 0.7 | 1.3 | 3.7 |
| FAP | 96.2 | 72.0 | 38.8 | 94.6 | 82.0 | 76.7 | 66.6 | 22.3 | 8.6 | 46.2 | 33.1 | 35.4 |
| **AdvMask (ours)** | 96.8 | 73.7 | 49.6 | 97.2 | 84.7 | **80.4** | 87.5 | 52.0 | 27.7 | 86.3 | 63.7 | **63.4** |

## C.4 ROBUSTNESS UNDER DIFFERENT VISION-LANGUAGE MODEL

In our experiments, we mainly use CLIP ViT as the image encoder, following previous works on VLM robustness (Zhou et al., 2024; Mao et al., 2023; Yu et al., 2024). However, since our AdvMask can apply binary masks to any modular components (e.g., self-attention, linear layers), it is architecture-agnostic and can generalize beyond CLIP-based models. To validate this, we conduct experiments on VisualBERT (Li et al., 2019), which processes image and text jointly through a BERT-style transformer. Specifically, we adopt AdvMask to VisualBERT on two multi-modal classification datasets (CrisisMMD2INF and CrisisMMD2HUM (Alam et al., 2018)). As shown in Table 15, the naive VisualBERT exhibits a substantial performance drop under adversarial attack, whereas our AdvMask significantly improves robustness without compromising clean performance. These results confirm that AdvMask generalizes beyond the CLIP ViT family and enhances practicality for broader VLM architectures.

| Dataset | Model | Clean | | Adv. | |
|---|---|---|---|---|---|
| | | Acc. | F1-score | Acc. | F1-score |
| CrisisMMD2INF | VisualBERT | 0.85 | 0.82 | 0.40 | 0.38 |
| | VisualBERT + AdvMask | **0.85** | **0.83** | **0.77** | **0.74** |
| CrisisMMD2HUM | VisualBERT | **0.78** | **0.68** | 0.12 | 0.07 |
| | VisualBERT + AdvMask | 0.77 | 0.65 | **0.59** | **0.48** |

Table 15: Results on VisualBERT architecture. We evaluate our AdvMask on VisualBERT by applying mask parameters to the self-attention layers of the last two encoder blocks of the model. For both naive and AdvMask-applied models, we perform 16-shot tuning on each of two different multi-modal classification datasets (i.e., CrisisMMD2INF and CrisisMMD2HUM datasets (Alam et al., 2018)). Adversarial training is conducted using PGD-2 ($\epsilon$=8/255, $\alpha$=1/255), and PGD-100 is used for evaluation.

## C.5 COMPATIBILIY WITH LEARNABLE PROMPT METHODS

Since AdvMask modifies only part of the visual encoder in VLMs, our method is orthogonal and complementary to prompt tuning techniques and can be flexibly integrated with them depending on task objectives. To demonstrate this, in Table 16, we present experiments combining AdvMask with CoOp (Zhou et al., 2022), a well-established learnable prompt tuning method. Specifically, we consider two cases: (1) combining independently trained CoOp prompts for the text encoder with AdvMask for the image encoder, and (2) further training learnable text prompts on top of the robust visual representations produced by AdvMask, allowing the prompts to adapt to robust features.

The results show that in case (1), simply combining our robust vision encoder with a learnable prompt yields significantly improved adversarial robustness compared to the original CLIP. This suggests that AdvMask strengthens the visual encoder's ability to generate robust representations, which can be effectively leveraged by any textual prompt. In case (2), adaptive prompt tuning further improves performance, as the contextual prompts are learned to align with the robust features extracted by the masked vision encoder. These findings indicate that AdvMask is not limited to fixed prompts and can be broadly applied alongside various prompt tuning strategies to enhance VLM robustness.

| Method | Clean Acc. | Adv. Acc. |
|---|---|---|
| CLIP | 56.6 | 4.8 |
| CLIP + CoOP | 71.1 | 15.8 |
| AdvMask + CoOP (case 1) | 58.5 | 37.7 |
| AdvMask + CoOP (case 2) | **66.3** | **44.7** |

Table 16: Integration of AdvMask with learnable prompt tuning method (i.e., CoOp). We report average clean and adversarial accuracy (%, ↑) on five downstream datasets in 16-shot setting. Two cases are compared: (1) combining independently trained CoOp prompts for the text encoder with AdvMask for the image encoder, and (2) further training learnable text prompts on top of the robust visual representations produced by AdvMask, allowing the prompts to adapt to robust features. Results averaged over 3 random trials.

## C.6 COMPARISON WITH FULLY FINE-TUNED BASELINE

In our main experiments, we focused on parameter-efficient adversarial tuning methods, as fully fine-tuning a large VLM is both computationally expensive and prone to overfitting in limited-data settings. Nevertheless, we agree that including a full fine-tuning baseline strengthens our claims. Following the reviewer's suggestion, we conducted full-parameter fine-tuning under the 16-shot setting, and the results are provided in Table 17. We find that while full fine-tuning achieves reasonable performance when enough samples are available (e.g., 16-shot), it performs poorly in low-data regimes (1-shot and 4-shot), exhibiting clear signs of overfitting. We also observed that full fine-tuning is highly sensitive to hyperparameters such as the learning rate, making it less stable under few-shot conditions. Importantly, the computational costs of the two approaches differ substantially: as shown in the table, full fine-tuning updates all parameters of the vision encoder, resulting in significantly higher training time and memory usage. In contrast, AdvMask learns only lightweight binary masks applied to a small subset of modules (i.e., MHSA layers), providing far greater efficiency while simultaneously achieving stronger adversarial robustness.

| Method | 1-shot | | 4-shots | | 16-shots | | Comp. Cost (train) | |
|---|---|---|---|---|---|---|---|---|
| | Clean | Adv. | Clean | Adv. | Clean | Adv. | Time (s) | Mem (MB) |
| CLIP | 56.6 | 4.8 | 56.6 | 4.8 | 56.6 | 4.8 | – | – |
| FAP | 28.6 | 9.3 | 54.0 | 26.0 | 64.3 | 40.2 | 0.73 | 2863 |
| Full FT | 31.0 | 10.4 | 43.5 | 21.7 | 67.7 | 43.9 | 0.85 | 7724 |
| AdvMask | 46.6 | 18.4 | 57.2 | 32.2 | 67.3 | 47.1 | 0.27 | 1581 |

Table 17: Comparison with fully fine-tuned baseline. We evaluate few-shot performance (1/4/16 shots) and training cost (time/memory per batch). Full fine-tuning (Full FT) updates all model parameters using adversarial training.

## C.7 APPLICABILITY BEYOND CLASSIFICATION

Since our method is designed to be readily applicable to a wide range of vision encoders, it is naturally extensible to a variety of visual-language tasks beyond classification. To support this, we test AdvMask on an image captioning task using LLaVA (Liu et al., 2023), a recent multimodal LLM that integrates a CLIP ViT-L/14 encoder with a large language model. Due to computational constraints, we kept LLaVA's projection layer (which maps visual tokens to the LLM input space) frozen, and replaced only the vision encoder with our AdvMask-tuned version (trained on ImageNet under the 16-shot setting). As shown in Table 18, although a drop in clean caption quality is observed, likely due to a distributional mismatch between AdvMask-tuned visual embeddings and the frozen

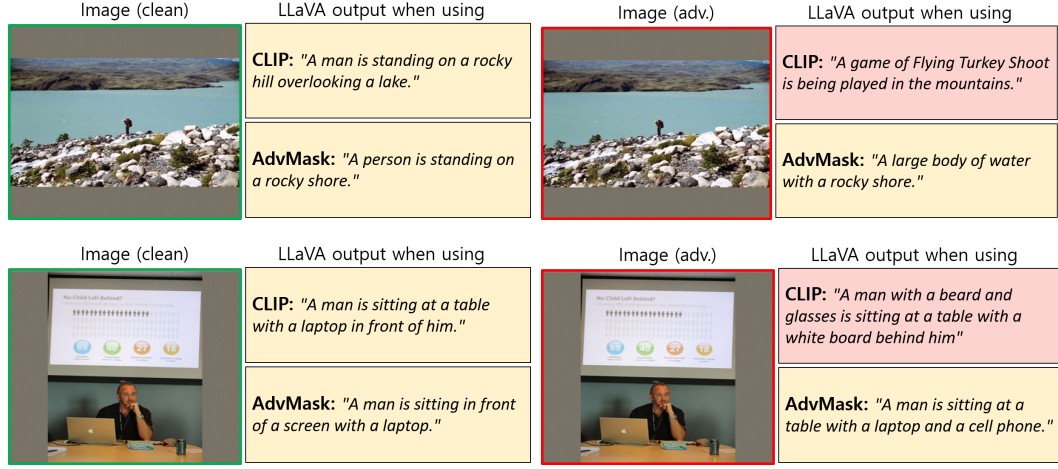

Figure 7: Visualization of LLaVA image captioning results on clean (left) and adversarial (right) examples.

projection layer, AdvMask still yields clear improvements in adversarial robustness even without any adaptation of the projection layer or the LLM. This demonstrates that AdvMask effectively suppresses perturbation-sensitive parameters at the vision-encoder level and can serve as a plug-and-play robustness module for downstream multimodal tasks. We believe these initial results support the promise of AdvMask as a task-agnostic robustness enhancer for VLMs.

Table 18: Robustness evaluation of AdvMask on LLaVA for image captioning task. We evaluate our AdvMask on the multimodal model (i.e., LLaVA), which integrates a CLIP ViT-L/14 image encoder and a Vicuna-7B language model. The task is image captioning on the Flickr30K dataset (500 samples). We report CIDEr scores (0-150, ↑) under clean and adversarial settings. Regarding attack settings, we use (i) a single-step APGD attack and (ii) a much stronger APGD-ensemble attack (i.e., multiple APGD variants at different precision levels) following Schlarmann et al. (2024a). AdvMask is tuned on ImageNet using a 16-shot setting, and applied without additional tuning to LLaVA's image encoder.

| Model | Clean | Adv. (APGD) | Adv. (Ensemble) |
|---|---|---|---|
| LLaVA (CLIP ViT-L/14) | 85.18 | 22.13 | 3.26 |
| LLaVA (CLIP ViT-L/14 + AdvMask) | 69.22 | 28.87 | 10.34 |

# D ADDITIONAL ANALYSIS

## D.1 ABLATION STUDY ON THE COEFFICIENT OF $\mathcal{L}_{\text{LAFA}}$

In Fig. 8, we present an ablation study on the coefficient $\lambda$ of the loss term $\mathcal{L}_{\text{LAFA}}$ in our objective function. This loss aims to align intermediate-layer features between clean and adversarial samples during tuning, enhancing representational robustness against adversarial attacks. The results show that our AdvMask consistently outperforms the competitive baseline (FAP), regardless of the coefficient setting. In our main experiments, we set $\lambda$ to 50.0, as excessively large coefficients (e.g., $\lambda = 100.0$) can slightly degrade clean accuracy due to overly constraining the feature space, particularly in the 16-shot scenario. Nevertheless, our AdvMask achieves competitive performance in both clean and adversarial accuracy through the proposed layer-wise adaptive feature alignment objective.

## D.2 ABLATION STUDY ON THE ADAPTIVE WEIGHTING SCHEME

One of the key contributions of our loss design is the adaptive weighting scheme in the LAFA loss (Sec. 2.3), which is particularly crucial for stabilizing mask tuning in few-shot scenarios. To validate its effectiveness, we provide an ablation study in Table 19, comparing performance with and without the adaptive weighting mechanism across different $\lambda$ values (i.e., the coefficient of the LAFA loss). The results show that incorporating adaptive weighting consistently outperforms the

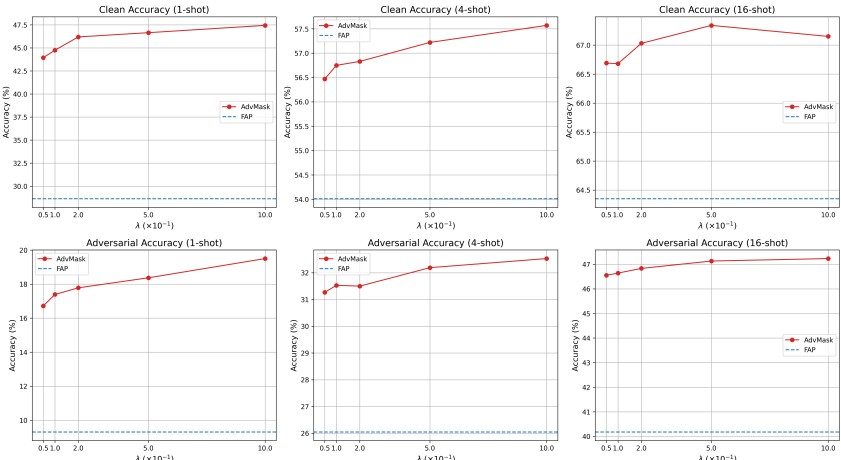

Figure 8: Ablation study on the coefficient $\lambda$ of $\mathcal{L}_{\text{LAFA}}$ in our loss function. We compare our method with FAP in the 1, 4, 16-shot setting with the same configurations in the main results.

unweighted counterpart, with the most significant gains observed in extremely low-shot settings (e.g., 1-shot). This is because the adaptive scheme emphasizes learning signals from more reliable samples, thereby reducing the risk of overfitting to noisy or misclassified examples, which is an especially important property under few-shot conditions. Regarding our adaptive weighting scheme in the early stage of tuning, although the model's initial confidence may not be perfectly reliable, the weight affects only the relative emphasis of each sample rather than removing its learning signal. Together with the warm-up strategy applied during the first epoch, this prevents early confidence errors from destabilizing optimization and enables LAFA to provide consistent gains across all shot settings.

| $\lambda$ | weighting | 1-shot | | 4-shots | | 16-shots | |
|---|---|---|---|---|---|---|---|
| | | Clean | Adv. | Clean | Adv. | Clean | Adv. |
| 10.0 | False | 42.0 | 16.4 | 56.1 | 30.7 | 66.5 | 46.3 |
| | True | **44.7** | **17.4** | **56.7** | **31.5** | **66.7** | **46.6** |
| 20.0 | False | 43.5 | 17.2 | 56.3 | 31.0 | 66.6 | 46.2 |
| | True | **46.2** | **17.8** | **56.8** | **31.5** | **67.0** | **46.8** |
| 50.0 | False | 44.5 | 17.8 | 56.6 | 32.1 | 66.9 | 46.8 |
| | True | **46.6** | **18.4** | **57.2** | **32.2** | **67.3** | **47.1** |

Table 19: Ablation study on the adaptive weighting scheme. We compare performance with and without adaptive weighting across different $\lambda$ values. We report clean and adversarial test accuracy (%, $\uparrow$) over 5 datasets in few-shot settings and results are averaged over 3 random trials.

## D.3 ABLATION STUDY ON MASK THRESHOLD ($\alpha$)

In our method, the mask threshold $\alpha$ controls the sparsity of the learned subnetwork by determining how aggressively real-valued masks are binarized. As shown in Table 20, increasing $\alpha$ (e.g., from 0.001 to 0.005) leads to higher sparsity and generally improves both clean and adversarial accuracy, since the learned mask better captures task-relevant and robust pathways while suppressing noise-vulnerable parameters. However, an excessively large $\alpha$ (e.g., 0.007) can slightly degrade performance due to over-pruning, which reduces the expressive power of the pre-trained network. Importantly, our AdvMask consistently outperforms the baseline in adversarial robustness across all $\alpha$ values, demonstrating its stability and effectiveness in balancing robustness and transferability.

| Method | 1-shot | | | 4-shots | | | 16-shots | | |
|---|---|---|---|---|---|---|---|---|---|
| | Clean | Adv. | Sparsity | Clean | Adv. | Sparsity | Clean | Adv. | Sparsity |
| FAP | 28.6 | 9.3 | – | 54.0 | 26.0 | – | 64.3 | 40.2 | – |
| AdvMask ($a$=0.001) | 44.8 | 15.3 | 0.01 | 53.8 | 29.1 | 0.03 | 65.5 | 44.0 | 0.12 |
| AdvMask ($a$=0.003) | 46.1 | 17.1 | 0.01 | 55.6 | 30.3 | 0.04 | 66.6 | 45.5 | 0.17 |
| AdvMask ($a$=0.005) | **46.6** | 18.4 | 0.02 | **57.2** | 32.2 | 0.06 | **67.3** | **47.1** | 0.27 |
| AdvMask ($a$=0.007) | 43.5 | **19.1** | 0.04 | 57.2 | **34.1** | 0.13 | 65.8 | 47.0 | 0.70 |

Table 20: Ablation study on the mask threshold $\alpha$. We report clean and adversarial test accuracy averaged over 5 datasets using 3 random trials under 1-shot, 4-shots, and 16-shots settings.

### D.4 ABLATION STUDY ON MASK INITIALIZATION

In all experiments, we initialize mask parameters with a constant value of $0.01$ and use a binarization threshold $\alpha = 0.005$, following Zheng et al. (2023). With this setup, all parameters start in the "on" state (i.e., identical to the original model), and during tuning, perturbation-vulnerable parameters are gradually pushed below the threshold and eventually deactivated through binarization. Due to this mechanism, the initialization value and the threshold ($\alpha$) are tightly coupled and jointly determine the sparsity of the learned mask (i.e., the proportion of deactivated parameters). An ablation study on the threshold is provided in Appendix Sec. D.3. In Fig. 21, we further provide an ablation study on the mask initialization value while fixing the threshold at $\alpha = 0.005$. Across all shot settings, we observe a clear pattern: larger initialization values (e.g., 0.02-0.05) lead to lower sparsity, since the mask values rarely fall below the threshold. However, excessively large initialization values often cause unstable tuning dynamics and degrade performance. Conversely, when the initialization value is small (e.g., 0.007), a larger number of informative parameters are inadvertently deactivated, resulting in a slight drop in clean accuracy. Despite these outcomes, initialization values around $0.01$ (the setting used in our main experiments) consistently achieve strong clean and adversarial performance, demonstrating stable behavior. Notably, these observations are consistent with the trends reported in Appendix Sec. D.3 regarding the effect of threshold $\alpha$.

| **Method** | **1-shot** | | | **4-shots** | | | **16-shots** | | |
|---|---|---|---|---|---|---|---|---|---|
| | Clean | Adv. | Sparsity | Clean | Adv. | Sparsity | Clean | Adv. | Sparsity |
| FAP | 28.6 | 9.3 | – | 54.0 | 26.0 | – | 64.3 | 40.2 | – |
| AdvMask (init=0.007) | 35.2 | 17.2 | 0.069 | 55.5 | **33.8** | 0.223 | 64.7 | 46.3 | 1.188 |
| AdvMask (init=0.01) | **46.6** | **18.4** | 0.017 | **57.2** | 32.2 | 0.062 | **67.3** | **47.1** | 0.273 |
| AdvMask (init=0.02) | 45.3 | 12.2 | 0.004 | 51.2 | 25.6 | 0.016 | 62.1 | 40.4 | 0.069 |
| AdvMask (init=0.05) | 45.9 | 5.0 | 0.001 | 47.2 | 15.8 | 0.005 | 53.1 | 29.4 | 0.019 |

,

Table 21: Ablation study on the mask initialization value (with $\alpha = 0.005$). We report clean and adversarial test accuracy averaged over 5 datasets using 3 random trials under the 1-shot, 4-shots, and 16-shots settings.

### D.5 ABLATION STUDY ON LAYER POSITIONS OF LAFA LOSS

In Table 22, we provide an ablation study on the layer positions where our LAFA loss is applied. Specifically, we divide the 12-layer encoder into four groups and compare performance when applying LAFA loss to each group (as well as to all groups). The results show that our AdvMask outperforms the competitive baseline (i.e., FAP) across all configurations, with stronger performance when applied to deeper or all layers. We believe this is because deactivating vulnerable parameters in later layers, which are closer to the model's final output, is more effective for improving robustness and adaptability. These results demonstrate that our approach is robust to hyperparameter choices and highlight the effectiveness of deactivating noise-sensitive parameters through layer-wise alignment.

| Method | 1-shot | | 4-shots | | 16-shots | |
|---|---|---|---|---|---|---|
| | Clean | Adv. | Clean | Adv. | Clean | Adv. |
| FAP | 28.6 | 9.3 | 54.0 | 26.0 | 64.3 | 40.2 |
| AdvMask ($l = \{0, 1, 2\}$) | 45.3 | 17.5 | 56.4 | 31.4 | 66.6 | 46.4 |
| AdvMask ($l = \{3, 4, 5\}$) | 45.9 | 17.6 | 56.2 | 31.2 | 66.6 | 46.5 |
| AdvMask ($l = \{6, 7, 8\}$) | 46.1 | 18.0 | 56.9 | 31.7 | 67.0 | 46.7 |
| AdvMask ($l = \{9, 10, 11\}$) | 44.8 | **19.2** | **57.7** | **32.8** | 66.8 | **47.3** |
| AdvMask ($l = $ all) | **46.6** | 18.4 | 57.2 | 32.2 | **67.3** | 47.1 |

Table 22: Ablation study on the layer positions where LAFA loss is applied. We report average test accuracy over 5 datasets in 1-shot, 4-shots, and 16-shots settings with 3 random trials.

### D.6 In-Depth Interpretation and Visualization of Learned Mask

**Mask Similarity Between Different Datasets.** To better understand how dataset characteristics influence the learned adversarial masks, we measure the similarity of masking patterns across dataset pairs. Specifically, we compute the overlap (IoU) over the deactivated parameters (i.e., positions where the mask value is 0), since these represent parameters identified as vulnerable to adversarial perturbations. As shown in Fig. 9, the mean IoU over the entire layers is relatively low, ranging from 0.075 to 0.124 (7.5%-12.4%) depending on the dataset pairs. This indicates that, globally, each dataset tends to highlight somewhat different parameter subsets as vulnerable. However, a layer-wise analysis reveals an interesting results. We observe that early layers consistently exhibit higher IoU than later layers, meaning that the overlap in masked positions is relatively larger near the input stage. This suggests that parameters in low-level feature extractors (i.e., closer to the input space) tend to be commonly vulnerable across datasets, leading AdvMask to deactivate a similar set of weights regardless of the dataset. In contrast, later layers show relatively lower IoU, indicating higher variability in which parameters are masked. These layers are more tightly coupled with downstream prediction behavior, and thus the masked parameters tend to reflect a combination of (1) perturbation-vulnerable weights and (2) dataset-specific parameters involved in task-level adaptation. As a result, the masking patterns diverge more noticeably across datasets in deeper layers. Overall, these findings highlight that our AdvMask captures both universal and dataset-dependent vulnerability structures within the model: early layers encode generalizable weak points shared across downstream datasets, while later layers reveal how vulnerability interacts with dataset-specific semantic alignment and adaptation.

**Mask Similarity Between Different Runs (i.e., Seeds).** We evaluate the similarity of the learned masks across different random seeds in Table 23. For each of the five datasets in the 16-shot setting, we train AdvMask using three independent runs with different seeds and quantify their similarity by measuring the IoU over deactivated parameters (i.e., positions where the mask equals 0). We focus on masked positions since the overall sparsity of the learned masks is extremely low ($\approx$0.27%), making IoU over activated parameters less informative (as most entries are equal to 1). For each dataset, we compute pairwise IoU across all seed pairs and report the averaged value as *Mean IoU over Masking Positions*. As shown in Table 23, despite differences in tuning samples and optimization trajectories, the learned masks exhibit a moderate level of overlap, with mean IoU values ranging from 0.20 to 0.31 (i.e., 20-31%). Importantly, both clean accuracy and adversarial robustness show only minor variance across seeds, indicating that the functional behavior of the model remains stable even when the exact masking locations differ. This is expected as the vast majority of parameters remain activated, resulting in highly similar feature extraction pathways, while the commonly deactivated parameters contribute to consistently improved robustness. Overall, these results suggest that AdvMask consistently identifies functionally similar sets of adversarially vulnerable parameters across different seeds. Although the precise masked parameters may vary due to the non-convex and combinatorial nature of mask optimization, the model reliably converges to masks that suppress semantically equivalent vulnerability patterns, leading to stable performance and robust behavior that is effectively independent of the random seed.

**Which Layers or Attention Heads Are Primarily Masked?** In Fig. 10, we present an analysis of which layers and attention heads are predominantly masked by AdvMask. Using the masks learned in the 16-shot setting for five different datasets, we compute sparsity (i.e., the percentage of deactivated

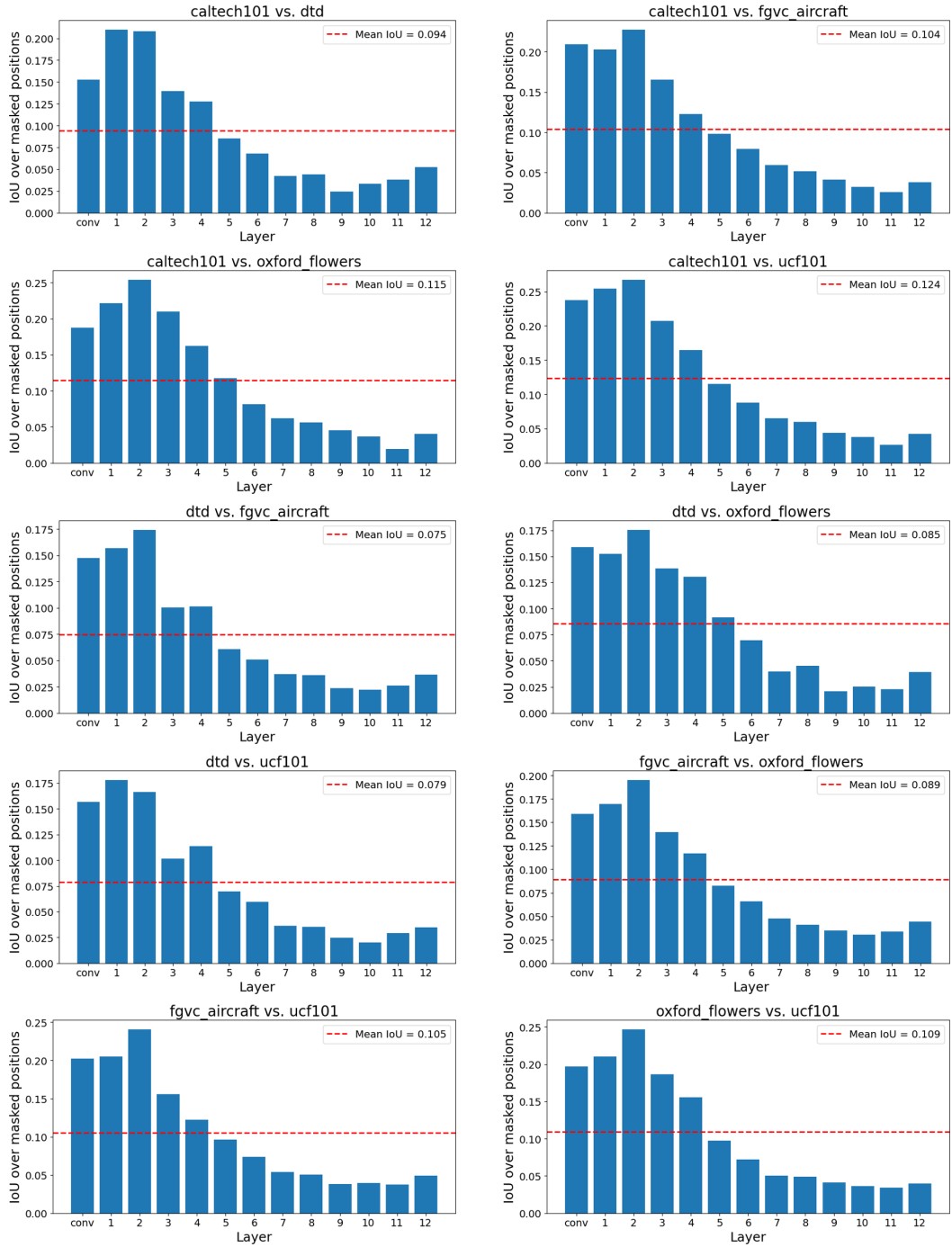

Figure 9: Layer-wise mask IoU across different dataset pairs in the 16-shot setting. Each subfigure shows the layer-wise similarity of masking positions between a pair of datasets, computed over the deactivated (masked) parameters. We use five datasets in total and report all pairwise combinations. The overall experimental setup and hyperparameters follow the main paper.

parameters) to quantify how aggressively each component is masked. To first examine which layers are more likely to be masked, Fig. 10a reports the layer-wise sparsity of the learned binary masks, averaged over the five datasets. We observe that sparsity consistently increases toward deeper layers, indicating that later layers (where representational shifts introduced by adversarial perturbations become more pronounced) play a more critical role in mask tuning. These layers require more

| Dataset | Mean IoU (%) | Clean Acc. | Adv. Acc. |
|---|---|---|---|
| Caltech101 | 0.25 | 92.9 ($\pm$0.05) | 75.8 ($\pm$0.21) |
| DTD | 0.20 | 58.4 ($\pm$0.17) | 35.5 ($\pm$1.23) |
| FGVCAircraft | 0.31 | 26.8 ($\pm$1.50) | 12.0 ($\pm$0.49) |
| OxfordFlowers | 0.31 | 88.0 ($\pm$0.61) | 69.9 ($\pm$0.29) |
| UCF101 | 0.22 | 70.6 ($\pm$0.86) | 42.4 ($\pm$0.29) |

Table 23: Mask similarity and performance stability across three independent runs (different seeds) in the 16-shot setting. We report the average IoU over deactivated parameters (masking positions), along with clean and adversarial accuracies (mean $\pm$ standard deviation).

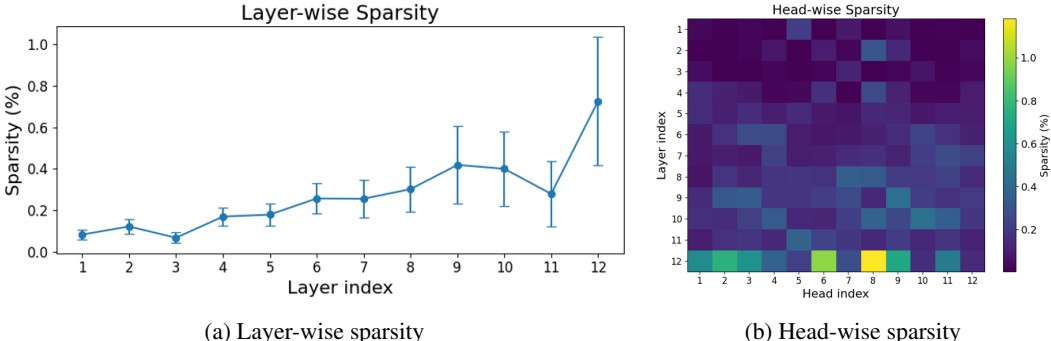

(a) Layer-wise sparsity            (b) Head-wise sparsity

Figure 10: Analysis of which components are predominantly masked by AdvMask: (a) layer-wise sparsity and (b) head-wise sparsity. Both results are computed in the 16-shot setting and averaged over five datasets.

extensive deactivation of vulnerable parameters to stabilize high-level features and maintain robust predictions for the downstream task. Another important observation is that the deeper layers, which are more closely tied to task-specific adaptation, exhibit higher variance across datasets since the degree and pattern of masking required for effective adaptation differs depending on the dataset. To further understand the masking behavior within the multi-head self-attention mechanism, Fig. 10b presents head-wise sparsity for each layer, averaged over the five datasets. Interestingly, certain heads exhibit consistently high sparsity across datasets; for example, the 6th and 8th heads in the final (12th) layer show particularly strong masking. This suggests that specific attention heads are universally prone to adversarial vulnerability, and suppressing them contributes disproportionately to the model's robustness. In other words, AdvMask systematically identifies and deactivates a small subset of structurally fragile heads that act as common failure points across datasets.

**Which Module Types Are Primarily Masked?** In Fig. 11, we present an analysis of which module types within the multi-head self-attention (MHSA) block are predominantly masked by AdvMask. Since our mask tuning is applied only to the MHSA components of each transformer block for both effectiveness and efficiency, we compare the sparsity of (1) the projection matrices responsible for generating Q, K, and V (denoted as `attn`), and (2) the output projection matrix (`attn.out_proj`), which maps the concatenated head outputs back to the model dimension. (See Sec. 3.3 for ablations demonstrating why MHSA layers are the most effective target for mask tuning.) As shown in Fig. 11a, both module types exhibit increasing sparsity toward deeper layers, reinforcing the observation that later transformer layers play a more influential role in robustness. The `attn.out_proj` module shows a notably sharper increase, suggesting that the integration stage of multi-head attention is particularly sensitive to adversarial vulnerabilities and thus requires stronger masking to stabilize high-level representations. Furthermore, Fig. 11b shows that `attn.out_proj` exhibits more than three times higher sparsity on average than `attn`. This may be attributed to its role in aggregating information from all attention heads (making it more susceptible to perturbed signals) as well as its tight connection to downstream task-specific adaptation. Consequently, suppressing vulnerable weights in this module helps prevent distorted signals from being propagated, improving both adversarial robustness and downstream adaptability.

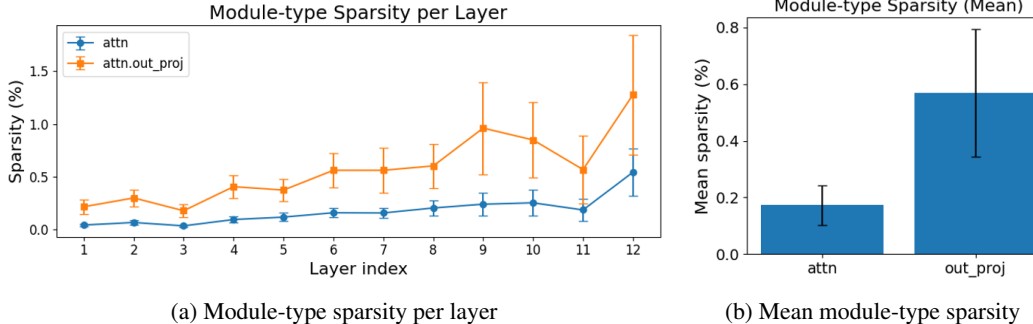

(a) Module-type sparsity per layer        (b) Mean module-type sparsity

Figure 11: Analysis of which module types within the multi-head self-attention block are predominantly masked by AdvMask. (a) Layer-wise sparsity for the `attn` and `attn.out_proj` modules. (b) Mean sparsity across layers, averaged over five datasets in the 16-shot setting.

### D.7 COMPUTATIONAL COST

Our method, AdvMask, is designed to be parameter-efficient by optimizing only a small set of binary mask parameters over a subset of the model (i.e., the self-attention layers, which account for  20% of the model), while keeping the rest of the pre-trained model frozen. To evaluate efficiency, we report quantitative comparisons of training and inference costs (latency and GPU memory usage per batch) in Table 24. The results show that (1) during training, AdvMask is more memory- and time-efficient than most baselines due to its lightweight mask tuning approach, and (2) during inference, although memory usage increases slightly from additional mask parameters, latency remains lower than or comparable to baselines, demonstrating practicality for deployment. Moreover, AdvMask is effective even in challenging few-shot settings, requiring only a small number of downstream samples, making it well-suited for resource-constrained scenarios. Overall, these results highlight that AdvMask offers practical advantages in both cost and data efficiency, particularly in few-shot scenarios.

| Method | Training | | Inference | |
|---|---|---|---|---|
| | Time (s) | Memory (MB) | Time (s) | Memory (MB) |
| CLIP | – | – | 0.05 | 1268 |
| AdvVP | 0.29 | 937 | 0.17 | 1561 |
| AdvVLP | 0.49 | 2789 | 0.15 | 1783 |
| AdvMaPLe | 0.40 | 1726 | 0.16 | 1809 |
| FAP | 0.73 | 2863 | 0.16 | 1809 |
| AdvMask (ours) | 0.27 | 1581 | 0.13 | 1946 |

Table 24: Computational cost for training and inference. We report detailed training and inference costs (i.e., time and memory usage per batch). All baselines use the same batch sizes (train: 4, test: 200), with adversarial sample generation cost included during training.

### D.8 EXTENSION OF ADVMASK BEYOND BINARY MASK

While our framework adopts binary masks to explicitly form selective neural pathways, the method can naturally be extended to soft-mask variants. To explore this direction, we implement a ternary version of AdvMask in which each mask element can take one of three values $\{0, 0.5, 1\}$. This is achieved by introducing two thresholds ($\alpha_1 = 0.005$, $\alpha_2 = 0.008$) while keeping all other configurations identical to the binary-mask setting. As shown in Table 25, the ternary mask achieves performance comparable to the binary version and, in certain cases (e.g., 4-shot adversarial accuracy), even slightly outperforms it due to its larger representational flexibility. Importantly, both binary and ternary variants consistently surpass baseline methods across different shot settings. These findings suggest that AdvMask can be readily extended to soft-mask formulations. However, binary masks offer significant practical advantages in terms of parameter compactness and deployability, as they require only a single bit per weight and are therefore highly efficient to store, transmit, and reuse. In

contrast, soft masks require higher-precision numerical values, increasing storage and deployment overhead. Empirically, we find that binary masking is sufficient to capture robust subnetworks while keeping computational cost efficient.

|  | 1-shot | | 4-shots | | 16-shots | |
|---|---|---|---|---|---|---|
| Method | Clean | Adv. | Clean | Adv. | Clean | Adv. |
| CLIP | 56.6 | 4.8 | 56.6 | 4.8 | 56.6 | 4.8 |
| FAP | 28.6 | 9.3 | 54.0 | 26.0 | 64.3 | 40.2 |
| AdvMask (binary) | **46.6** | 18.4 | 57.2 | 32.2 | **67.3** | **47.1** |
| AdvMask (ternary) | 44.8 | **18.8** | **58.9** | **35.3** | 66.4 | 46.9 |

Table 25: Comparison between binary and ternary masks. Clean and adversarial test accuracy averaged over 5 datasets using 3 random trials under the 1-shot, 4-shots, and 16-shots settings.

### D.9 PER-CLASS PERFORMANCE ANALYSIS

In this section, we perform a comprehensive per-class performance analysis to identify which categories benefit the most from AdvMask. We compute class-wise adversarial accuracies for both CLIP and AdvMask across all 101 categories on Caltech101 dataset. Out findings show that AdvMask significantly improves robustness across the majority of categories, with the largest gains appearing in categories that are highly brittle under adversarial perturbations. Over 40 categories where CLIP completely fails (0% accuracy), AdvMask substantially recovers performance, often reaching 40-80% accuracy. To clearly highlight which categories benefit the most, we include a summary in Table 26, showing the Top-10 and Bottom-10 classes by adversarial accuracy improvement. This table directly illustrates that AdvMask yields the largest benefits for the most adversarially fragile categories. For examples, several categories exhibit extreme improvements, such as "car side", "cellphone", "okapi", "face", "ferry", "dalmatian", "tick", "grand piano", and "barrel", where accuracy improves by +0.7 to +1.0 absolute points (i.e., 70-80%). These classes typically rely on high-frequency or texture-sensitive cues, which are severely corrupted by adversarial perturbations. AdvMask effectively suppresses unstable activations, allowing the model to retain semantically meaningful features. This per-class analysis supports our main claim: AdvMask selectively strengthens robustness for categories most vulnerable to adversarial noise, while maintaining strong performance on clean samples.

## E DISCUSSIONS

### E.1 CONCEPTUAL DISTINCTION BETWEEN ADVERSARIAL FINE-TUNING AND ADVMASK

Adversarial fine-tuning and our proposed AdvMask share the high-level goal of improving robustness, but they operate through fundamentally different mechanisms. Standard adversarial fine-tuning directly updates the pretrained weights, altering the internal representations of the model to fit the downstream task. Such weight modifications often overwrite or distort the pretrained feature space, a phenomenon described in prior work (Schlarmann et al., 2024b), and may degrade generalization on unseen tasks. In contrast, AdvMask preserves all pretrained parameters and instead learns binary on/off gating that selectively suppresses perturbation-sensitive units. This design identifies a robust subnetwork embedded within the original VLM while maintaining its generalizable pretraining knowledge. Unlike weight fine-tuning, AdvMask does not modify or overwrite representations; it merely routes computation through more robust pathways. AdvMask is also computationally more efficient. The method updates only lightweight binary masks applied to a subset of modules (primarily MHSA layers, roughly 20% of VLM parameters) rather than optimizing the full set of model weights. Empirically, we find that this targeted gating is sufficient to form robust neural pathways that improve adversarial robustness while retaining the strong zero-shot and few-shot generalization ability of the underlying model.

Table 26: Per-class adversarial accuracy improvements (0.0∼1.0) of AdvMask over CLIP. Top-10 classes show the largest positive improvements, while Bottom-10 show the smallest improvements. $\Delta = \text{Acc}_{\text{AdvMask}} - \text{Acc}_{\text{CLIP}}$.

| Class | CLIP | AdvMask | $\Delta$ |
|---|---|---|---|
| **Top-10 Improved Classes** | | | |
| car_side | 0.000 | 1.000 | +1.000 |
| face | 0.015 | 0.954 | +0.939 |
| cellphone | 0.118 | 1.000 | +0.882 |
| okapi | 0.091 | 1.000 | +0.909 |
| ferry | 0.050 | 0.950 | +0.900 |
| accordion | 0.125 | 1.000 | +0.875 |
| barrel | 0.071 | 1.000 | +0.929 |
| dalmatian | 0.250 | 0.950 | +0.700 |
| grand_piano | 0.276 | 0.966 | +0.690 |
| tick | 0.333 | 1.000 | +0.667 |
| **Bottom-10 Improved Classes** | | | |
| crocodile_head | 0.000 | 0.067 | +0.067 |
| platypus | 0.000 | 0.100 | +0.100 |
| crayfish | 0.000 | 0.143 | +0.143 |
| crab | 0.000 | 0.227 | +0.227 |
| scorpion | 0.000 | 0.240 | +0.240 |
| mayfly | 0.000 | 0.250 | +0.250 |
| crocodile | 0.000 | 0.267 | +0.267 |
| bass | 0.062 | 0.375 | +0.313 |
| lotus | 0.000 | 0.350 | +0.350 |
| rhino | 0.000 | 0.353 | +0.353 |

# F  LIMITATIONS

In our implementation, to achieve computational efficiency during adversarial mask tuning, AdvMask selectively optimizes mask parameters in multi-head self-attention (MHSA) layers. However, this approach may leave other layers potentially vulnerable to adversarial attacks. Although we demonstrate in Sec. 3.3 that masking MHSA layers is indeed more effective for adversarial robustness compared to masking MLP layers (in terms of both efficiency and performance), it remains possible that even more selective or adaptive masking strategies could further enhance robustness. Therefore, identifying additional or alternative layers and adaptively tuning masks (while maintaining efficiency) could be an important direction for future research.

