# OpenReview forum: "Identifying Robust Neural Pathways: Few-Shot Adversarial Mask Tuning for Vision-Language Models"
_ICLR.cc/2026/Conference — ICLR 2026 Poster_

### Official Review · Reviewer_SmjY · 2025-10-25

**Soundness:** 3
**Presentation:** 4
**Contribution:** 3
**Rating:** 6
**Confidence:** 4

**Summary:**

This paper introduces AdvMask to improve the adversarial robustness of in few-shot settings. Instead of fine-tuning model weights, it learns binary masks to identify and activate robust neural pathways, preserving stable features under attack. The proposed LAFA loss further enhances robustness by aligning intermediate features between clean and adversarial samples.

**Strengths:**

1. The setting of the task of this work is clear. The convert from Mask Tuning to Adv Mask Tuning is interesting and theoretical.

2. The method framework is clearly introduced, while using visualization to help readers quickly understand the method.

3. The paper provides sufficient experimental evidence and further insight.

**Weaknesses:**

1. This work involves two-shot scenarios: few-shot training and zero-shot evaluation. The authors are advised to clearly explain and distinguish these in the introduction to facilitate understanding. For example, the phrase "overfitting in a few-shot setting" on line 54 and "zero-shot robustness" on line 55 may not be aligned settings, but their use together could easily lead to misunderstanding and confusion.

2. In the ablations Table 17, Table 18, etc., the 16-shot performances of "47.1" and "47.3" seem to be different from the "41.99" given in Table 10. How are they obtained?

3. If the author's training data uses 3.2% of the data, then is the training time also reduced to 3.2% compared to TGA-ZSR? I didn't see a direct comparison in the paper.

**Questions:**

1. Are there any visualization or statistical results that show the specific situation of the final mask, and can we summarize in a regular way which weights are more important for adversarial and which need to be ignored?

2. I'm curious if this few-shot approach would be applicable to the scenarios tuned for LVLM in FARE? Are there any challenges in doing so?

[1] Schlarmann C, Singh N D, Croce F, et al. Robust clip: Unsupervised adversarial fine-tuning of vision embeddings for robust large vision-language models[J]. arXiv preprint arXiv:2402.12336, 2024.

---

> ### Author Response · Authors · 2025-11-21
>
> Dear Reviewer ```SmjY```,
> Thank you for the reviewer's constructive feedback and for acknowledging the contributions of our work. We have carefully reviewed all of the comments and provide detailed responses to each point below. If any further clarification or discussion is needed, we would be glad to elaborate.
>
> &nbsp;
>
> > ## **W1. Clarification on settings in introduction section**
>
> We appreciate the reviewer for pointing out the potential confusion between the two setups. We agree that these are distinct evaluation settings, and using the terminology without clear differentiation may lead to misunderstandings. In the revised manuscript, **we explicitly clarify the distinction**: few-shot tuning refers to scenarios where each downstream dataset provides only a small number of labeled samples for adaptation, whereas zero-shot robustness refers to evaluating robustness without any downstream samples, relying instead on a held-out source dataset. We have updated the relevant statements (lines 54-58) to clearly reflect this distinction.
>
> &nbsp;
>
> > ## **W2. Clarification on scores reported**
>
> We clarify that **the scores in Table 10 and those in Tables 17-18 are produced under different experimental settings.** Specifically, Table 10 corresponds to the zero-shot robustness setting, where no samples from the target downstream datasets are provided. Instead, the model is tuned only on 16-shot samples from a held-out source dataset (i.e., TinyImageNet) and evaluated zero-shot on the unseen downstream datasets. In contrast, Tables 17 and 18 report few-shot ablation results, where each downstream dataset provides 16-shot samples for task-specific adaptation. Since the tuning data and evaluation conditions are different between zero-shot and few-shot scenarios, the resulting performance values are not directly comparable. We apologize for the confusion and have updated the captions of Table 10 and Table 11 in the revised manuscript to clearly indicate the corresponding experimental settings.
>
> &nbsp;
>
> > ## **W3. Comparison of training time with respect to training data**
>
> We thank the reviewer for this valuable question. In our zero-shot robustness setup (Table 2 of main paper), TGA-ZSR is tuned on the entire source dataset (100%), whereas our AdvMask uses only 3.2% (16-shot) of the same source data. As the reviewer correctly noted, if batch size and the number of epochs are the same, **the total training time is approximately proportional to the number of tuning samples; thus, using only 3.2% of the data naturally reduces the overall wall-clock training time.**
> However, a more meaningful comparison is the per-batch training efficiency, since total training time depends heavily on dataset size. We note that TGA-ZSR updates all model weights during adversarial training, which is computationally expensive for large VLMs. In contrast, AdvMask trains only lightweight binary masks over a subset of modules, making each optimization step far more efficient. To support this, we report **per-batch computational cost** in Table 1 below. Under identical batch size, **AdvMask requires substantially less time and GPU memory per batch compared to TGA-ZSR.** This confirms that AdvMask is not only more data-efficient but also more computationally efficient in practice.
>
> - **Table 1. Per-batch training time and memory usage comparison between TGA-ZSR and AdvMask**
> : We report training costs (i.e., time and memory usage per batch), using batch size of 4.
>
> | Method      | Time (s/batch) | Memory (MB) |
> | ----------- | -------------- | ----------- |
> | **TGA-ZSR (full finetuning)** | 0.85           | 7724        |
> | **AdvMask** | 0.27           | 1581        |

---

> ### Author Response · Authors · 2025-11-21
>
> > ## **Q1. Visualization and statistics on learned mask**
>
> Thank you for the insightful suggestion. To address this point, we provide **a layer-wise CKA analysis in Fig. 4 of Sec. 3.3, illustrating how the learned mask shapes a more stable and robust neural pathway.** Furthermore, during the rebuttal period, we expanded our interpretability and visualization results and included detailed analyses in **Appendix Sec. D.6.**
>
> As suggested by the reviewer, we present comprehensive statistics of the learned masks to reveal which parts of the model are most frequently deactivated. First, in Fig. 9 of Sec. D.6 (and summarized in Tables 2 and 3 below), we analyze **which layers and attention heads are primarily masked.** These results consistently show that deeper transformer layers and specific attention heads (particularly in the final layer) exhibit significantly higher sparsity, indicating that AdvMask identifies structural components that are most susceptible to adversarial vulnerabilities.
>
> - **Table 2. Layer-wise Sparsity**
> : Layer-wise sparsity (%) of the learned mask under the 16-shot setting. Detailed settings and results are provided in Fig. 9(a) of Appendix Sec. D.6.
>
> | Layer        | L1   | L2   | L4   | L6   | L8   | L10  | L12  |
> | ------------ | ---- | ---- | ---- | ---- | ---- | ---- | ---- |
> | Sparsity (%) | 0.08 | 0.12 | 0.17 | 0.26 | 0.30 | 0.40 | 0.73 |
>
> &nbsp;
>
> - **Table 3. Head-wise Sparsity per Layer**
> : Head-wise sparsity (%) for selected layers (1, 6, and 12). Detailed settings and results are provided in Fig. 9(b) of Appendix Sec. D.6.
>
> | Layer \ Head  | 1    | 2    | 3    | 4    | 5    | 6        | 7    | 8        | 9    | 10   | 11   | 12   |
> | ---------------- | ---- | ---- | ---- | ---- | ---- | -------- | ---- | -------- | ---- | ---- | ---- | ---- |
> | **Layer 1**      | 0.03 | 0.01 | 0.02 | 0.01 | 0.22 | 0.00     | 0.09 | 0.01     | 0.06 | 0.01 | 0.01 | 0.01 |
> | **Layer 6**      | 0.09 | 0.17 | 0.28 | 0.27 | 0.09 | 0.08     | 0.09 | 0.11     | 0.16 | 0.24 | 0.17 | 0.11 |
> | **Layer 12**     | 0.57 | 0.76 | 0.60 | 0.37 | 0.23 | **0.98** | 0.29 | **1.18** | 0.71 | 0.16 | 0.49 | 0.15 |
>
> &nbsp;
>
> Also, we analyze **which module types within the multi-head self-attention (MHSA) block are predominantly masked**, as shown in Fig. 10 and summarized in Table 4 below. Specifically, we compare the sparsity of (1) the projection matrices responsible for generating Q, K, and V (denoted as 'attn'), and (2) the output projection matrix ('attn.out_proj'), which maps the concatenated multi-head outputs back to the model dimension.
> These results show that 'attn.out_proj' exhibits more than three times higher sparsity than the Q/K/V projection matrices. This indicates that the aggregation stage of multi-head attention is both more exposed to adversarial distortions and more tightly tied to downstream task-specific adaptation. Accordingly, AdvMask suppresses weights in this module to prevent corrupted signals from propagating into deeper layers.
>
> - **Table 4. Module-type Sparsity**
> : Module-type sparsity (%) of the learned mask. Detailed results are provided in Fig. 10 of Appendix Sec. D.6.
>
> | Layer        | attn | attn.out_proj |
> | ------------ | ---- | ------------- |
> | Sparsity (%) | 0.17 | 0.57          |

---

> ### Author Response · Authors · 2025-11-21
>
> > ## **Q2. Applicability to LVLM in FARE-like scenarios**
>
> We appreciate the reviewer for this constructive question. From a conceptual standpoint, **AdvMask is fully compatible with LVLM pipelines, as it operates exclusively on the vision encoder without modifying downstream text encoders or language models.** This makes AdvMask a plug-and-play robustness module, but with a different objective (few-shot robust adaptation rather than preserving zero-shot embeddings).
>
> To provide preliminary evidence, **we applied AdvMask to LLaVA for an image-captioning task like in FARE [1].** Due to computational constraints, we froze the entire multimodal architecture (including the vision-to-LLM projection layer) and replaced only the vision encoder (i.e., CLIP ViT-L/14) with its AdvMask-tuned counterpart (trained on ImageNet under the 16-shot setting). As summarized in Table 5 below, we observe a drop in clean caption quality, likely due to a distributional mismatch between AdvMask-tuned visual embeddings and the frozen projection layer (similar to the “embedding drift’’ issue discussed in FARE). Nevertheless, **AdvMask yields clear improvements in adversarial robustness even without any adaptation of the projection layer or the LLM.** This demonstrates that AdvMask can enhance robustness at the vision-encoder level in LVLMs, despite the absence of downstream alignment. We believe that integrating AdvMask with a jointly tuned projection layer (as done in FARE) would further mitigate representation mismatch, and we view such LVLM extensions of AdvMask as a promising future direction. We will include this experiment and discussion in the revised manuscript.
>
> - **Table 5. Robustness evaluation of AdvMask on LLaVA for image captioning task**
> : We evaluate our AdvMask on the multimodal model (i.e., LLaVA), which integrates a CLIP ViT-L/14 image encoder and a Vicuna-7B language model. The task is image captioning on Flickr30K dataset (500 samples). We report CIDEr scores (0–150, ↑) under clean and adversarial settings. AdvMask is tuned on ImageNet using a 16-shot setting, and applied without additional tuning to LLaVA’s image encoder.
>
> |                                 | Clean | Adv   |
> | ------------------------------- | ----- | ----- |
> | LLaVA (CLIP ViT-L/14)           | 85.18 | 22.13 |
> | LLaVA (CLIP ViT-L/14 + **AdvMask**) | 69.22 | 28.87 |
>
>
> [1] Schlarmann et al., Robust clip: Unsupervised adversarial fine-tuning of vision embeddings for robust large vision-language models. ICML'24.

---

### Official Review · Reviewer_kPP8 · 2025-10-30

**Soundness:** 2
**Presentation:** 3
**Contribution:** 2
**Rating:** 4
**Confidence:** 4

**Summary:**

The paper investigates an interesting adversarial defense mechanism without explicitly tuning the model weights; instead, the proposed method optimizes a model weight-level mask across different layers to enhance robustness. In other words, the proposed method shields potentially vulnerable weights to adversarial robustness. Furthermore, the paper introduced a layer-wise adaptive feature alignment scheme to optimize such a model weight-level mask by aligning clean features and their adversarial counterparts in the few-shot setup. Experiments across diverse datasets and scenarios demonstrate the generalization capability of the proposed method. Further ablations justifies the efficacy of the design of the proposed pathway method.

**Strengths:**

1. The proposed idea is interesting. Instead of tuning the whole weight map of VLMs to improve robustness, the paper explores an alternative way by exploring the weight-level mask to remove some implicitly vulnerable model weights against adversarial attacks.
2. The paper is well-written and organized. A detailed recap of previous works and a background introduction are given.
3. Extensive experiments across diverse benchmarks and scenarios are provided. In addition, a series of ablation analyses is given to verify the effectiveness of each module. Insights in Figure 4 are also interesting.

**Weaknesses:**

1. The proposed method might not be novel in the context of adversarial learning (for both single-modal and multimodal architectures). [a] has already explored the model weights connected with adversarial robustness. Although [a] is based on a single-modal architecture, its idea can also be easily extended to a multimodal backbone.

2. The evaluated adversarial attacks are primarily low-intensity (low perturbation radius) attacks with eps=1/255. It's questionable that if the proposed method also exhibits robustness against stronger adversarial attacks with higher eps (e.g., 4/255, 8/255)

3. Can the mask be regarded as part of the weights of VLMs? If so, I think that finetuning the VLM weights can also achieve the same effect, the mask would be mostly like some scailing of the standard adversarial finetuning, which means the proposed mask is an indirect adversarial finetuning. Then, it should achieve similar performance compared with adversarial finetunin.

4. It seems that the mask is only for the image encoder. But the paper focuses on VLMs. In this case, it would be more appropriate to consider both branches, otherwise the work would be mostly similar to single-modal adversarial learning works.

[a] Improving Generalization of Adversarial Training via Robust Critical Fine-Tuning (ICCV 2023)

**Questions:**

1. Can the authors evaluate the text-level (BERT-Attack) or joint image-text-level attacks (CO-Attack) in addition to image-level attacks only?

2. In addition to image classification, VLMs are powerful in diverse visual-language tasks, e.g., image captioning, Visual question answering. Can the authors test some of them instead of classification?

3. Can the proposed mask be a soft format instead of the 0-1 style?

[b] BERT-ATTACK: Adversarial Attack Against BERT Using BERT (EMNLP-2020)

[c] Towards Adversarial Attack on Vision-Language Pre-training Models (ACMMM 2022)

---

> ### Author Response · Authors · 2025-11-21
>
> Dear Reviewer ```kPP8```,
> We are truly grateful for the reviewer's thorough feedback and valuable insights. We have carefully addressed each of points in the following responses. If there are any additional concerns you would like us to clarify, we would greatly appreciate the opportunity to respond further.
>
> &nbsp;
>
> > ## **W1. Novelty in terms of adversarial learning**
>
> We appreciate the reviewer for pointing us to RiFT [1] which provides valuable insights. While we acknowledge the conceptual connection that both RiFT and AdvMask explore robustness-related structure within neural networks, we would like to clarify that the two approaches differ fundamentally in their motivation, optimization procedure, and problem setting.
>
> (1) **Objectives.** RiFT aims to improve the **generalization performance of an adversarially trained model** by identifying non-robust-critical modules (via Module Robust Criticality, MRC) and fine-tuning those modules using clean data. In contrast, our goal is to enhance the **adversarial robustness of a pretrained model** in a few-shot setting without access to adversarially trained models and without modifying any pretrained weights. Thus, the goals and settings of the two methods are fundamentally different.
>
> (2) **Optimization protocols.** In RiFT, the model first takes weight perturbation to compute MRC; subsequently, the identified modules are fine-tuned, and an interpolation between adversarially trained weights and fine-tuned weights is performed. AdvMask instead adopts a parameter-efficient mask tuning strategy: all pretrained weights remain frozen, and only binary masks are optimized using the proposed LAFA loss to suppress perturbation-vulnerable pathways. **No fine-tuning or weight interpolation is involved, making the adaptation procedure substantially lighter and more suited for few-shot VLM robustness.**
>
> (3) **Applicability to VLMs.** While RiFT could be extended to multimodal models, we respectfully point out that **applying RiFT in our VLM setting would require a fully adversarially trained VLM backbone** (typically infeasible given the scale and cost of VLM pretraining). Furthermore, computing MRC through weight perturbations on such large models would incur substantial computational overhead. AdvMask avoids these challenges by introducing, to the best of our knowledge, the first mask-based robust adaptation framework for pretrained VLMs in few-shot scenarios.
>
> In summary, although both works share a related intuition regarding robustness variation across model components, they are designed for different purposes, operate in different regimes, and rely on distinct optimization mechanisms. We clarified these distinctions in the revised related work section (lines 500-503).
>
> [1] Zhu et al., Improving Generalization of Adversarial Training via Robust Critical Fine-Tuning. ICCV'23.
>
> &nbsp;
>
> > ## **W2. Ablation study on attack strength**
>
> Thank you for the valuable suggestion. In Appendix Sec. C.1, we have evaluated **AdvMask under stronger adversarial settings, including higher perturbation bounds.** As shown in Fig. 5 (or Table 1 below), AdvMask consistently maintains strong adversarial robustness across multiple perturbation levels ($\epsilon$ = 1/255, 2/255, 4/255), demonstrating that its effectiveness is not limited to low-intensity attacks. These results confirm that AdvMask provides reliable robustness even against stronger adversarial attacks.
>
> - **Table 1. Robustness under Different Perturbation Bounds**
> : We evaluate the robustness of different perturbation bounds ($\epsilon$ = 1/255, 2/255, 4/255). Experimental details are provided in Sec. C.1 of Appendix.
>
> | Adv. Acc.      | $\epsilon$=1/255            | $\epsilon$=2/255            | $\epsilon$=4/255           |
> | ----------- | -------------- | -------------- | ------------- |
> | **CLIP**    | 2.66           | 0.18           | 0.01          |
> | **FAP**     | 35.78 (± 1.10) | 11.08 (± 0.33) | 1.57 (± 0.32) |
> | **AdvMask** | **39.95 (± 0.22)** | **23.22 (± 0.14)** | **7.00 (± 0.20)** |

---

> ### Author Response · Authors · 2025-11-21
>
> > ## **W3. Is mask tuning indirect weight fine-tuning?**
>
> We appreciate the reviewer’s interesting perspective. Although our AdvMask may appear similar to adversarial fine-tuning in that it modulates the effect of weights, AdvMask operates through a fundamentally different mechanism. Standard adversarial fine-tuning updates the pretrained weights themselves, **altering the model’s internal representations to fit the downstream task.** In contrast, **AdvMask applies binary on/off gating to the pretrained parameters without modifying any weights, identifying a robust subnetwork within the original VLM** while preserving its generalizable pretrained knowledge.
>
> Prior work [1] has also shown that **adversarial weight updates can distort pretrained feature spaces and degrade performance on unseen tasks**; by comparison, AdvMask avoids such representation drift by selectively deactivating only perturbation-sensitive units while keeping the rest of the feature extractor intact. Empirically, we find that this targeted gating is sufficient to form robust neural pathways while retaining the strong generalization ability of the underlying VLM. We included this conceptual distinction and discussions in Sec. E.1 of the revised manuscript.
>
> [1] Schlarmann et al., Robust clip: Unsupervised adversarial fine-tuning of vision embeddings for robust large vision-language models. ICML'24.
>
> &nbsp;
>
> > ## **W4. Consideration of both modalities**
>
> Thank you for bringing up this important point. We would like to emphasize that prior works on VLM robustness have predominantly focused on the prompt branch, while several recent studies (e.g., [2]) highlight that the image encoder is the primary source of adversarial vulnerability. One of our core contributions is **demonstrating that applying mask-based parameter suppression to the visual encoder achieves substantially stronger adversarial robustness for VLMs.**
>
> Furthermore, **AdvMask is orthogonal and complementary to prompt-tuning approaches and can be flexibly integrated depending on task objectives.** To verify this, we provided additional experiments in Appendix Sec. C.5 (summarized in Table 2 below) by combining AdvMask with CoOp [1], a representative learnable prompt tuning method. Specifically, we evaluated (1) a simple combination of independently trained CoOp prompts with our AdvMask-tuned image encoder, and (2) CoOp prompts further trained on top of the robust visual representations produced by AdvMask. As a result, both settings improve adversarial robustness over CLIP, and the latter achieved even stronger performance as the prompts adapted to the stable and robust features from our masked encoder. These results confirm that **AdvMask is not limited to the image branch; instead, it serves as a complementary module that can be integrated with various prompt-side adaptation techniques.**
>
> - **Table 2. Integration of AdvMask with learnable prompt tuning method (CoOp)**
> : We report average clean and adversarial accuracy (%, ↑) in 16-shot setting when combining AdvMask with CoOp. Two cases are compared: (1) independently trained CoOp prompts combined with AdvMask-tuned vision encoder, and (2) CoOp prompts further adapted to robust visual features extracted via AdvMask.
>
> |                       | Clean Acc. | Adv. Acc. |
> | --------------------- | ---------- | --------- |
> | CLIP                  | 56.6       | 4.8       |
> | CLIP+CoOP             | **71.1**       | 15.8      |
> | AdvMask+CoOP (case 1) | 58.5       | 37.7      |
> | AdvMask+CoOP (case 2) | 66.3       | **44.7**      |
>
> [1] Zhou et al., Learning to Prompt for Vision-Language Models, IJCV'22.
> [2] Schlarmann et al., Robust clip: Unsupervised adversarial fine-tuning of vision embeddings for robust large vision-language models. ICML'24.

---

> ### Author Response · Authors · 2025-11-21
>
> > ## **Q1. Evaluation under text- or joint-level attacks (BERT-Attack, CO-Attack)**
>
> Following the reviewer's comment, we conducted **additional experiments to evaluate robustness against text-level and joint image-text-level adversarial attacks.** Specifically, we assess whether the learned masks (although trained only with image-level adversarial supervision) can still provide robustness when different modalities are further attacked. We consider two evaluation settings beyond standard image-level PGD attacks: (1) Independent multimodal attacks, where PGD is applied to the image while BERT-Attack [1] perturbs the text prompts; (2) Joint multimodal attacks, following CoAttack [2], where image and text embeddings are perturbed in a coordinated manner within the joint multimodal space. As shown in Table 3 below, performance decreases under stronger multimodal attack scenarios, but **our AdvMask still maintains robustness even though it was never trained with text or joint-level perturbations.** These results suggest that our mask-based robustness generalizes beyond image-level perturbations. We included these results and discussion in Appendix Sec. C.2 of the revised manuscript.
>
> - **Table 3. Robustness under multimodal adversarial attacks**
> : By using the masks trained with 16-shot downstream samples, we assess whether the learned masks can still provide robustness when different modalities are attacked. We report average adversarial accuracy (%, ↑) over 5 datasets across three different runs.
>
> | **Attack Type**                                 | **Adv. Accuracy (%)** |
> | ----------------------------------------------- | ---------------- |
> | CLIP (baseline on CoAttack)                       | 7.3              |
> | PGD + BERT-Attack (independent)                 | 29.2             |
> | PGD + BERT-Attack (CoAttack) | 28.5             |
>
> [1] Li et al., Bert-attack: Adversarial attack against bert using bert. EMNLP'20.
> [2] Zhang et al., Towards adversarial attack on vision-language pre-training models. ACM-ICM'22.
>
> &nbsp;
>
> > ## **Q2. Applicability beyond classification**
>
> We appreciate for the constructive suggestion. As our method is designed to be readily applicable to a wide range of vision encoders, it is naturally extensible to a variety of visual-language tasks beyond classification. To provide preliminary evidence, **we tested AdvMask on an image captioning task using LLaVA**, a recent multimodal LLM that integrates a CLIP ViT-L/14 encoder with a large language model. Due to computational constraints, we kept LLaVA’s projection layer (which maps visual tokens to the LLM input space) frozen, and replaced only the vision encoder with our AdvMask-tuned version (trained on ImageNet under the 16-shot setting).
>
> As summarized in Table 4 below, although a drop in clean caption quality is observed, likely due to a distributional mismatch between AdvMask-tuned visual embeddings and the frozen projection layer, AdvMask still yields **clear improvements in adversarial robustness even without any adaptation of the projection layer or the LLM.** This demonstrates that AdvMask effectively suppresses perturbation-sensitive parameters at the vision-encoder level and can serve as a plug-and-play robustness module for downstream multimodal tasks. We believe these initial results support the promise of AdvMask as a task-agnostic robustness enhancer for VLMs, and will incorporate these findings and discussions in the revised manuscript.
>
> - **Table 4. Robustness evaluation of AdvMask on LLaVA for image captioning task**
> : We evaluate our AdvMask on the multimodal model (i.e., LLaVA), which integrates a CLIP ViT-L/14 image encoder and a Vicuna-7B language model. The task is image captioning on Flickr30K dataset (500 samples). We report CIDEr scores (0-150, ↑) under clean and adversarial settings. AdvMask is tuned on ImageNet using a 16-shot setting, and applied without additional tuning to LLaVA’s image encoder.
>
> |                                 | Clean | Adv   |
> | ------------------------------- | ----- | ----- |
> | LLaVA (CLIP ViT-L/14)           | 85.18 | 22.13 |
> | LLaVA (CLIP ViT-L/14 + **AdvMask**) | 69.22 | 28.87 |

---

> > ### Comment · Reviewer_kPP8 · 2025-11-27
> >
> > Many thanks for the rebuttal. I still have several concerns:
> >
> > 1. It seems that LLaVA suffers from a significant drop in clean accuracy. It also seems that the conclusion is different from the FARE paper: why does the clean LLaVA have quite a good adv accuracy? It should be vulnerable to adversarial examples, right?
> >
> > 2. Can the authors provide visualisations for the LLaVA captioning?
> >
> > 3. How about other tasks evaluated also in FARE (VQA, hallucination, and Science QA)
> >
> > 4. "Prior work [1] has also shown that adversarial weight updates can distort pretrained feature spaces and degrade performance on unseen tasks;"  Could you provide the original text for this claim? I find it hard to find this in the FARE paper.

---

> > > ### Author Response · Authors · 2025-12-01
> > >
> > > Dear Reviewer ```kPP8```,
> > > We sincerly thank for the follow-up comments. We address each concern in detail below and supplement them in Sec. C.7 of our revised manuscript.
> > >
> > > &nbsp;
> > >
> > > > **Q1.** It seems that LLaVA suffers from a significant drop in clean accuracy. It also seems that the conclusion is different from the FARE paper: why does the clean LLaVA have quite a good adv accuracy? It should be vulnerable to adversarial examples, right?
> > >
> > > **A1.** The drop in clean accuracy is consistent with the behavior reported in the FARE paper and is likely caused by representation shift, as adversarially training the CLIP encoder naturally alters its feature distribution. Also, regarding the adversarial accuracy of the vanilla LLaVA model, **the relatively higher robustness is due to the difference in attack settings.** Specifically, we initially applied a single-step APGD attack for preliminary experimentation, whereas the FARE paper evaluates robustness under a much stronger APGD-ensemble attack (i.e., multiple APGD variants at different precision levels).
> > > To resolve this discrepancy, we now provide results using the same APGD-ensemble setting as FARE (see Table 1 below). These updated results confirm that AdvMask consistently improves robustness when integrated into the LLaVA vision encoder, even under stronger adversarial attacks.
> > >
> > > - **Table 1. Robustness evaluation of AdvMask on LLaVA for image captioning task**
> > >
> > > |                                 | Clean | Adv (APGD)| Adv (APGD-Ensemble)|
> > > | ------------------------------- | ----- | ----- | ----- |
> > > | LLaVA (CLIP ViT-L/14)           | 85.18 | 22.13 | 3.26 |
> > > | LLaVA (CLIP ViT-L/14 + **AdvMask**) | 69.22 | 28.87 | 10.34 |
> > >
> > > &nbsp;
> > >
> > > > **Q2.** Can the authors provide visualisations for the LLaVA captioning?
> > >
> > > **A2.** Following the reviewer's suggestion, we provide **qualitative visualizations of captioning outputs in Fig. 7 of Appendix Sec. C.4.** These examples illustrate that the baseline LLaVA often produces hallucinated or totally irrelevant captions under adversarial perturbations, whereas the AdvMask-enhanced model generates captions that remain faithful to the visual content even under attack.
> > >
> > > &nbsp;
> > >
> > > > **Q3.** How about other tasks evaluated also in FARE (VQA, hallucination, and Science QA)
> > >
> > > **A3.** In Table 2 below, we additionally evaluate our AdvMask-enhanced vision encoder on the **TextVQA dataset**, following the evaluation protocol used in FARE. We would like to clarify that the fundamental objectives of FARE and AdvMask are different: FARE aims to improve zero-shot robustness by adversarially training on an auxiliary dataset, whereas our AdvMask is designed to enhance robustness during few-shot adaptation on a targeted downstream task. Nevertheless, the results show that **AdvMask still provides meaningful robustness gains in this new multimodal task setting**, indicating that our approach can generalize beyond image classification. We believe that even stronger improvements are possible when explicitly optimizing for zero-shot robustness (e.g., preserving the original feature distribution), which we leave as a promising direction for future work.
> > >
> > >
> > > - **Table 2. Robustness evaluation of AdvMask on LLaVA for visual question-answering (VQA) task**
> > >
> > > |                                 | Clean | Adv (APGD)| Adv (APGD-Ensemble)|
> > > | ------------------------------- | ----- | ----- | ----- |
> > > | LLaVA (CLIP ViT-L/14)           | 43.5 | 7.4 | 0.0 |
> > > | LLaVA (CLIP ViT-L/14 + **AdvMask**) | 28.6 | 15.1 | 10.8 |
> > >
> > > &nbsp;
> > >
> > > > **Q4.** "Prior work [1] has also shown that adversarial weight updates can distort pretrained feature spaces and degrade performance on unseen tasks;" Could you provide the original text for this claim? I find it hard to find this in the FARE paper.
> > >
> > > **A4.** The statement refers to findings reported in the FARE paper regarding TeCoA-style supervised adversarial fine-tuning, which updates all weights of the CLIP encoder. **Such weight updates significantly reshape the pretrained embedding space, resulting in degraded zero-shot performance on unseen tasks.** For clarity, the original passages from the FARE paper are as follows: (1) “the fine-tuning can lead to heavy distortions with respect to unseen classes, which explains the high losses in standard performance for other down-stream zero-shot classification tasks.” (Sec. 3.2), and (2) “TeCoA-loss does not aim to preserve the original CLIP embedding and thus can introduce arbitrary distortions, which causes the degradation of performance in zero-shot classification and other down-stream tasks.” (Appendix Sec. C.4).

---

> ### Author Response · Authors · 2025-11-21
>
> > ## **Q3. Soft mask instead of binary mask**
>
> Thank you for the interesting question. While our framework uses binary masks to explicitly form selective neural pathways, our proposed method can be extended to soft mask formats. To simply explore this, we implemented **a ternary version of AdvMask, where each mask element takes one of three values {0, 1/2, 1}.** This was achieved by introducing two thresholds (e.g., $\alpha_1 = 0.005$, $\alpha_2 = 0.008$), while keeping all other settings identical to the original configuration. As shown in Table 5 below, the ternary mask achieves performance comparable to the binary version and, in some cases (e.g., 4-shot adversarial accuracy), slightly outperforms it due to having a larger representational flexibility. Importantly, both binary and ternary versions consistently outperform baseline methods across all shot settings.
>
> These results indicate that AdvMask can be naturally extended to soft-mask variants. However, **binary masks offer practical advantages in parameter compactness and deployability, as they require only a single bit per weight and thus are extremely efficient to store and transfer.** Empirically, we found the binary mask sufficient to capture robust subnetworks while keeping computational cost efficient. We have incorporated these results and a discussion of potential soft-mask extensions in Sec. D.8 of the revised manuscript.
>
> - **Table 5. Comparison between binary and ternary masks**
> : Comparison between binary and ternary versions of AdvMask. Each mask element takes one of three values {0, 1/2, 1} by introducing two thresholds ($\alpha_1 = 0.005$, $\alpha_2 = 0.008$), while keeping all other settings identical to the original configuration. Clean and adversarial accuracy (%) averaged over 5 datasets using 3 random trials.
>
> | Method                | Clean (1-shot) | Adv. (1-shot) | Clean (4-shots) | Adv. (4-shots) | Clean (16-shots) | Adv. (16-shots) |
> | --------------------- | -------------- | ------------- | --------------- | -------------- | ---------------- | --------------- |
> | **CLIP**              | 56.6           | 4.8           | 56.6            | 4.8            | 56.6             | 4.8             |
> | **FAP**               | 28.6           | 9.3           | 54.0            | 26.0           | 64.3             | 40.2            |
> | **AdvMask (binary)**  | **46.6**       | 18.4          | 57.2            | 32.2           | **67.3**         | **47.1**        |
> | **AdvMask (ternary)** | 44.8           | **18.8**      | **58.9**        | **35.3**       | 66.4             | 46.9            |

---

### Official Review · Reviewer_RUtB · 2025-10-31

**Soundness:** 3
**Presentation:** 3
**Contribution:** 3
**Rating:** 6
**Confidence:** 4

**Summary:**

This paper proposes AdvMask, a method to enhance adversarial robustness of Vision-Language Models (VLMs) in few-shot settings. Instead of modifying pre-trained weights or learning prompts, AdvMask learns binary masks that selectively deactivate parameters vulnerable to adversarial perturbations, effectively identifying robust neural pathways within the vision encoder. The authors introduce a new loss (LAFA) that aligns intermediate features between clean and adversarial samples with confidence-based weighting. Experiments on 11 datasets show AdvMask outperforms prompt-based baselines (AdvVP, AdvVLP, AdvMaPLe, FAP) in few-shot adversarial robustness while maintaining competitive clean accuracy.

**Strengths:**

. The concept of finding robust neural pathways via binary masks is creative and well-motivated.

.The framing as deactivating vulnerable parameters rather than adding robust features is a fresh perspective on adversarial defense.

. The experimental evaluation is thorough.

. Substantial improvements over baselines across most datasets.

.Clean accuracy recovers with more shots.

**Weaknesses:**

. The model is trained with only 2 steps of PGD at a very small noise level (ε = 1/255) and tested mostly at the same level. This is weak to prove robustness and may hide gradient masking. The authors should test with stronger attacks (for example ε = 4/255) to make sure the binary mask and straight-through estimator do not block gradients.

. The adaptive weight in LAFA uses the model’s own prediction confidence. At early stages, this confidence can be wrong, which may make the model ignore “hard but useful” samples. The authors could try using a teacher model or stop the gradient from this weight to prevent bias.

. The paper claims that certain layers or heads are more robust, but there is no clear visualization. It would help to show which layers or attention heads are most often masked and whether this pattern is consistent across datasets or random seeds.

. The paper argues that full fine-tuning overfits in few-shot cases, but this is not shown. A simple 16-shot full fine-tuning baseline would make this claim stronger.

. The paper says the method is efficient, but Table 19 shows higher inference memory than some baselines. A simple memory breakdown would help clarify this.

. Modern papers show that some defenses look strong until the attack is adapted to the defense itself. Here, the main defensive component is the mask and LAFA, so it would be good to test with attacks that target them directly.

**Questions:**

1. Can you show a baseline where the model is fully fine-tuned with 16 samples per class?

2. Which layers or heads are masked the most? Are these patterns stable across runs?

3. Why did you choose the mask initialization values?

4. Where does the method fail? For example, why is performance lower on Cars, Food101, and Aircraft datasets?

5. Why is the inference memory higher than other lightweight methods?

6. How does sparsity (number of masked weights) relate to robustness?

7. Can you add stronger attacks (ε = 4/255)?

8. Have you tested an adaptive attack that specifically targets the mask or LAFA?


---- Additional Suggestions

. Add a baseline where the mask is trained without adversarial samples to see if adversarial tuning is truly necessary.

. Show how the mask changes during training (sparsity per epoch).

. Include a small table showing scaling to a larger backbone such as ViT-L/14.

. Report per-class accuracy to show which categories benefit the most.

---

> ### Author Response · Authors · 2025-11-21
>
> Dear Reviewer ```RUtB```,
> We appreciate the reviewer for taking the time to provide thoughtful and constructive feedback on our work. We carefully reviewed all comments you raised and addressed each of them in detail in the responses below. If you have any remaining concerns to discuss any aspect further, we would be happy to continue the conversation.
>
> &nbsp;
>
> > ## **W1, Q7. Testing with stronger perturbations**
>
> We agree that evaluating under stronger perturbations is important to support our claims. **To address this, we have included the results using stronger adversarial attacks ($\epsilon$ = 2/255 and $\epsilon$ = 4/255) in Appendix Sec. C.1.** As shown in Fig. 5 (and Table 1 below), AdvMask maintains stable and competitive robustness across all perturbation strengths, while strong baseline such as FAP experience a sharp degradation as $\epsilon$ increases. Importantly, AdvMask continues to improve robustness even at $\epsilon$ = 4/255, confirming that the binary mask mechanism and straight-through estimator do not cause gradient obfuscation. These findings demonstrate that AdvMask provides reliable robustness beyond the perturbation level.
>
> - **Table 1. Robustness under Different Perturbation Bounds**
> : We evaluate the robustness of different perturbation bounds ($\epsilon$ = 1/255, 2/255, 4/255). Experimental details are provided in Sec. C.1 of Appendix.
>
> | Adv. Acc.      | $\epsilon$=1/255            | $\epsilon$=2/255            | $\epsilon$=4/255           |
> | ----------- | -------------- | -------------- | ------------- |
> | **CLIP**    | 2.66           | 0.18           | 0.01          |
> | **FAP**     | 35.78 (± 1.10) | 11.08 (± 0.33) | 1.57 (± 0.32) |
> | **AdvMask** | **39.95 (± 0.22)** | **23.22 (± 0.14)** | **7.00 (± 0.20)** |
>
> &nbsp;
>
>
> > ## **W2. Adaptive weighing in early-stage of tuning**
>
> We appreciate the reviewer’s insightful suggestion. As the reviewer pointed out, early-stage confidence is indeed an important consideration for our adaptive weighting design in LAFA. **Our choice is motivated by the fact that the pretrained CLIP model, due to its strong generalization capability, produces reasonably reliable confidence estimates on clean samples even before adaptation.** Empirically, we find that this confidence-based weighting effectively suppresses noisy gradients in few-shot regimes and contributes to stable mask tuning across different shot settings (Appendix Sec. D.2).
>
> Importantly, **even when the model’s initial prediction is incorrect, the adaptive weight only modulates the relative contribution of the sample rather than discarding its learning signal.** In practice, this does not significantly alter the learning dynamics in few-shot settings, as demonstrated by our empirical observations. Moreover, we apply **a warm-up strategy during the first epoch for all methods, which further mitigates instability arising from early-stage confidence errors**. We agree that incorporating a teacher model or stopping gradients through the adaptive weight is a promising enhancement that may further stabilize early-stage optimization; we plan to explore these directions in future work. We have clarified these points in Sec. D.2 of the revised manuscript.

---

> ### Author Response · Authors · 2025-11-21
>
> > ## **W3, Q2. Visualization and interpretability of learned mask**
>
> We appreciate the reviewer’s suggestion for clearer visualizations and interpretations. In response, we added comprehensive analyses of which layers and attention heads are masked, together with their consistency across datasets and random seeds (we provied detailed results on Appendix Sec. D.6).
>
> First, to investigate **which layers or attention heads are primarily masked**, we report layer-wise sparsity in Fig. 9(a), with a summarized version provided in Table 2 below. We observe that **deeper layers consistently exhibit higher sparsity, indicating that robustness relies more heavily on suppressing vulnerabilities in later layers.** This is likely because later layers encode more high-level, task-relevant information, and adversarial perturbations accumulate as they propagate through the network, resulting in larger representation shifts that require stronger masking.
>
> - **Table 2. Layer-wise Sparsity**
> : Layer-wise sparsity (%) of the learned mask under the 16-shot setting. Detailed settings and results are provided in Fig. 9(a) of Appendix Sec. D.6.
>
> | Layer        | L1   | L2   | L4   | L6   | L8   | L10  | L12  |
> | ------------ | ---- | ---- | ---- | ---- | ---- | ---- | ---- |
> | Sparsity (%) | 0.08 | 0.12 | 0.17 | 0.26 | 0.30 | 0.40 | 0.73 |
>
> &nbsp;
>
> Next, Fig. 9(b) reports **head-wise sparsity for each layer** (see Table 3 below for a summary), showing that **certain attention heads (e.g., heads 6 and 8 in the final layer) are consistently masked across datasets.** This suggests that AdvMask reliably identifies structurally fragile heads that tend to be universally vulnerable and act as common failure points for robust downstream prediction. In other words, even though different datasets have different semantics, there exist specific heads that systematically amplify adversarial noise, and AdvMask deactivates them accordingly.
>
> - **Table 3. Head-wise Sparsity per Layer**
> : Head-wise sparsity (%) for selected layers (1, 6, and 12). Detailed settings and results are provided in Fig. 9(b) of Appendix Sec. D.6.
>
> | Layer \ Head  | 1    | 2    | 3    | 4    | 5    | 6        | 7    | 8        | 9    | 10   | 11   | 12   |
> | ---------------- | ---- | ---- | ---- | ---- | ---- | -------- | ---- | -------- | ---- | ---- | ---- | ---- |
> | **Layer 1**      | 0.03 | 0.01 | 0.02 | 0.01 | 0.22 | 0.00     | 0.09 | 0.01     | 0.06 | 0.01 | 0.01 | 0.01 |
> | **Layer 6**      | 0.09 | 0.17 | 0.28 | 0.27 | 0.09 | 0.08     | 0.09 | 0.11     | 0.16 | 0.24 | 0.17 | 0.11 |
> | **Layer 12**     | 0.57 | 0.76 | 0.60 | 0.37 | 0.23 | **0.98** | 0.29 | **1.18** | 0.71 | 0.16 | 0.49 | 0.15 |
>
> &nbsp;
>
> We also analyze **seed-level consistency by reporting the IoU of masked parameters across three independent runs** for each dataset (see Table 19 in Appendix Sec. D.5 or Table 4 below). Despite variations in tuning samples and optimization trajectories caused by different seeds, the learned masks exhibit moderate overlap, with mean IoU values ranging from 0.20 to 0.31 (20-31%). More importantly, clean and robust accuracy remain highly stable across seeds, showing that AdvMask converges to functionally equivalent masking patterns even when the exact masked locations differ. This consistency indicates that the method reliably suppresses semantically similar vulnerability pathways and supports robust downstream adaptation. In addition, for a more thorough interpretation, Appendix Sec. D.6 also includes analyses on mask similarity across datasets and on which module types (e.g., Q/K/V projections vs. output projection) are most frequently masked. Thank you again for the constructive feedback.
>
> - **Table 4. Mask Similarity and Performance Stability Across Seeds**
> : Mean IoU over masked parameters and corresponding clean/robust accuracies across three random seeds for each dataset under the 16-shot setting. Detailed settings are provided in Table 19 of Appendix Sec. D.6.
>
> | Dataset       | Mean IoU (%) | Clean Acc. (± std) | Robust Acc. (± std) |
> | ------------- | ---------------------- | ------------------ | ------------------- |
> | Caltech101    | 0.25                   | 92.9 (± 0.05)        | 75.8 (± 0.21)         |
> | DTD           | 0.20                   | 58.4 (± 0.17)        | 35.5 (± 1.23)         |
> | FGVCAircraft  | 0.31                   | 26.8 (± 1.50)        | 12.0 (± 0.49)         |
> | OxfordFlowers | 0.31                   | 88.0 (± 0.61)        | 69.9 (± 0.29)         |
> | UCF101        | 0.22                   | 70.6 (± 0.86)        | 42.4 (± 0.29)         |

---

> ### Author Response · Authors · 2025-11-21
>
> > ## **W4, Q1. Full fine-tuning baseline**
>
> Thank you for this valuable suggestion. In our main experiments, we focused on parameter-efficient adversarial tuning methods, as fully fine-tuning a large VLM is both computationally expensive and prone to overfitting in limited-data settings. Nevertheless, we agree that including a full fine-tuning baseline strengthens our claims. Following the reviewer’s suggestion, we conducted full-parameter fine-tuning under the 16-shot setting, and the results are provided in Table 5 below. We find that while full fine-tuning achieves reasonable performance when enough samples are available (e.g., 16-shot), **it performs poorly in low-data regimes (1-shot and 4-shot), exhibiting clear signs of overfitting.**
>
> Importantly, the computational costs of the two approaches differ substantially: as shown in the table, **full fine-tuning updates all parameters of the vision encoder, resulting in significantly higher training time and memory usage.** In contrast, AdvMask learns only lightweight binary masks applied to a small subset of modules (i.e., MHSA layers), providing far greater efficiency while simultaneously achieving stronger adversarial robustness. We have included this comparison and the related discussion in Appendix Sec. C.6 of the revised manuscript.
>
> - **Table 5. Comparison with fully fine-tuned baseline**
> : We evaluate few-shot performance (1/4/16 shots) and training cost (time/memory per batch) per batch. Full fine-tuning (Full FT) updates all model parameters using adversarial training.
>
> | Method              | 1-shot Clean | 1-shot Adv | 4-shots Clean | 4-shots Adv | 16-shots Clean | 16-shots Adv | Train Time (s) | Memory (MB) |
> |-|-|-|-|-|-|-|-|-|
> | CLIP                | 56.6         | 4.8        | 56.6          | 4.8         | 56.6           | 4.8          | –          | –           |
> | FAP                 | 28.6         | 9.3        | 54.0          | 26.0        | 64.3           | 40.2         | 0.73       | 2863        |
> | Full FT             | 31.0         | 10.4       | 43.5          | 21.7        | 67.7           | 43.9         | 0.85       | 7724        |
> | AdvMask             | 46.6         | 18.4       | 57.2          | 32.2        | 67.3           | 47.1         | 0.27       | 1581        |
>
> &nbsp;
>
> > ## **W5, Q5. Efficiency and memory usage**
>
> We appreciate the reviewer for raising this important point. As reviewer noted, AdvMask exhibits slightly higher inference memory than some lightweight baselines. We clarify that **this increase is expected and comes from the storage of the learned binary masks applied to the multi-head self-attention layers.** Although this mask adds a small amount of additional memory, it is extremely compact as each mask entry requires only a single bit, and the total mask size is negligible compared to the full VLM parameters. Importantly, this small memory overhead does not affect test-time latency; **our inference speed remains faster than other baselines** since the binary mask introduces only a simple elementwise gating operation with no additional computation. Moreover, the overall efficiency benefits of AdvMask arise during training: only partial mask parameters are optimized while all pretrained weights remain frozen, yielding **significantly lower training-time memory usage and faster optimization** than the other methods. Therefore, we believe that AdvMask can be a practical choice for resource-constrained or few-shot robustness adaptation scenarios.
>
> &nbsp;
>
> > ## **W6. Q8. Adaptive attack robustness**
>
> We appreciate the reviewer’s thoughtful question on adaptive attacks. In fact, **all adversarial evaluations in our paper already operate under a fully adaptive setting.** During testing, the adversary computes perturbations by backpropagating through the defended masked model $f_{\theta \odot M_{bin}}$, not the original CLIP encoder. This means that the attacker has complete access to the learned binary mask, the straight-through estimator used during binarization, and the LAFA-tuned pathways. Consequently, **the generated perturbation explicitly targets the mask and the LAFA mechanism themselves**, rather than exploiting gradients of the unmasked model.
>
> Under this setting, gradients for perturbations are computed after the binary mask is applied, so the attacker adapts to exactly the same computation graph that the defense relies upon. Despite being exposed to such an adaptive attack, our defended model consistently preserves strong adversarial robustness across datasets and shot settings. This suggests that **the effectiveness of AdvMask arises from genuinely suppressing perturbation-vulnerable parameters and inducing inherently robust feature pathways.** We clarified this evaluation details in the revised manuscript (Sec. 3.1, line 312-314).

---

> ### Author Response · Authors · 2025-11-21
>
> > ## **Q3. Mask initialization values**
>
> In all experiments, we initialize mask parameters with a small positive constant ($1\times10^{-2}$) while using a binarization threshold of $5\times10^{-3}$, following prior work [1]. This setup ensures that **all parameters begin in the “on’’ state, and only perturbation-vulnerable parameters are gradually pushed below the threshold and deactivated during mask tuning.**
>
> To further understand the effect of initialization, we conducted an ablation study, with results provided in Table 6 below. We observe the following trends: (1) when the initialization value is relatively small (e.g., 0.007), the mask becomes overly aggressive, causing informative parameters to be deactivated and leading to a slight drop in clean accuracy; (2) when the initialization value is large (e.g., 0.02-0.05), the model exhibits unstable tuning dynamics and degraded performance. In contrast, initialization values near 0.01 consistently achieve strong clean and adversarial performance. It tends to keep sufficient parameters active at the start while still allowing the mask to selectively suppress perturbation-vulnerable parameters. Additional discussions are provided in Appendix Sec. D.4.
>
> - **Table 6. Ablation study on the mask initialization value**
> : Clean and adversarial accuracy averaged over 5 datasets (3 trials) under 16-shot setting. The mask threshold is fixed at $\alpha=0.005$. Detailed discussions and results are provided in Appendix Sec. D.4.
>
> | Method                     | Clean (16-shots) | Adv. (16-shots) | Sparsity |
> | - | - | - | - |
> | **FAP**                    | 64.3             | 40.2            | —        |
> | **AdvMask (init = 0.007)** | 64.7             | 46.3            | 1.188    |
> | **AdvMask (init = 0.01)**  | **67.3**         | **47.1**        | 0.273    |
> | **AdvMask (init = 0.02)**  | 62.1             | 40.4            | 0.069    |
> | **AdvMask (init = 0.05)**  | 53.1             | 29.4            | 0.019    |
>
> [1] Zheng et al., Regularized mask tuning: Uncovering hidden knowledge in pre-trained vision-language models. ICCV'23.
>
>
> &nbsp;
>
>
> > ## **Q4. Failure cases of AdvMask**
>
> As the reviewer pointed out, while AdvMask achieves strong robustness improvements on most datasets, the performance gains naturally vary depending on the intrinsic characteristics of each dataset. The datasets mentioned require relatively fine-grained visual discrimination, where class boundaries depend on subtle, high-frequency cues (e.g., texture details, part-level contours, and small structural variations). **Our mask-based approach intentionally suppresses perturbation-sensitive parameters, and some of these vulnerable units reside in high-frequency feature channels that are important for fine-grained recognition.** As a result, AdvMask may inadvertently deactivate features that are brittle yet informative for these domains, leading to smaller robustness improvements compared to datasets that rely more on coarse, semantic information. Nevertheless, across the majority of downstream datasets, AdvMask consistently outperforms baseline methods, indicating that its robustness benefits outweigh such edge cases.
>
>
> &nbsp;
>
>
> > ## **Q6. Sparsity vs. Robustness**
>
> We thank the reviewer for raising this important question. In Appendix Sec. D.3, we explicitly analyze the relationship between mask sparsity and robustness, where sparsity is controlled through the mask threshold $\alpha$. As summarized in Table 17 (and Table 7 below), **increasing $\alpha$ results in higher sparsity (i.e., more perturbation-vulnerable parameters are deactivated), and this generally leads to improved adversarial robustness.** This trend supports our claim that suppressing fragile parameters helps reveal a more stable and attack-resistant subnetwork within the pretrained model.
>
> However, we also observe that an **excessively large $\alpha$ (e.g., 0.007) can cause over-pruning**: although robustness may remain high, clean accuracy can slightly decrease due to the reduced expressive capacity of the resulting subnetwork. Overall, AdvMask demonstrates stable and competitive robustness across all sparsity levels, indicating that the method naturally balances robustness and transferability over a broad range of $\alpha$ values.
>
> - **Table 7. Sparsity (controlled by mask threshold ($\alpha$)) vs. Robustness**
> : We report 16-shot test accuracy (%, ↑) averaged over 5 datasets using 3 random trials.
>
> | Method                 | Clean (16-shots) | Adv (16-shots) | Sparsity |
> |-| --|------|-----|
> | FAP                    | 64.3              | 40.2            | –        |
> | AdvMask ($\alpha$ = 0.001)| 65.5              | 44.0            | 0.12     |
> | AdvMask ($\alpha$ = 0.003)| 66.6              | 45.5            | 0.17     |
> | AdvMask ($\alpha$ = 0.005)| **67.3**              | **47.1**            | 0.27     |
> | AdvMask ($\alpha$ = 0.007)| 65.8              | 47.0            | 0.70     |

---

> ### Author Response · Authors · 2025-11-21
>
> > ## **S1. Non-adversarial mask tuning baseline**
>
> Following the suggestion, **we evaluated a baseline that trains the mask only on clean samples using a clean cross-entropy (CE) loss, without any adversarial samples.** The results are provided in Table 8 below. As expected, this baseline finds pathways optimized purely for downstream adaptability, which leads to an improvement in clean accuracy. However, its adversarial robustness remains poor (7.2%). In contrast, our AdvMask explicitly incorporates adversarial samples into the tuning process, enabling it to discover pathways that are not only task-adapted but also resistant to perturbations. As a result, AdvMask significantly outperforms in adversarial robustness while maintaining strong clean accuracy, confirming that adversarial supervision is essential for learning robust subnetworks.
>
> - **Table 8. Comparision to non-adversarial mask tuning baseline**
> : We report 16-shot test accuracy (%, ↑) averaged over 5 datasets using 3 random trials.
>
> | Method              | Clean | Adv  |
> |---------------------|-------|------|
> | CLIP                | 56.6  | 4.8  |
> | Clean CE loss (no adv tuning) | 69.1  | 7.2  |
> | AdvMask (ours)      | 67.3  | 47.1 |
>
> &nbsp;
>
> > ## **S2. How the mask changes during tuning?**
>
> As training progresses, perturbation-vulnerable parameters are gradually pushed below the threshold and are deactivated through binarization. Following the reviewer’s suggestion, we report how mask sparsity evolves over training epochs in Table 9 below using an exemplar dataset (DTD). As expected, **sparsity increases monotonically as training proceeds, reflecting the progressive identification and suppression of vulnerable parameters, and stabilizes as the mask converges.** This confirms that AdvMask gradually forms a robust subnetwork over the course of tuning.
>
> - **Table 9. Sparsity per epoch**
> : We report sparsity per epoch in 16-shot AdvMask tuning on exemplar dataset (DTD).
>
> | Epoch | 1     | 2     | 3     | 4     | 5     | 6     | 7     | 8     | 9     | 10    |
> |-------|-------|-------|-------|-------|-------|-------|-------|-------|-------|-------|
> | Sparsity (%) | 0.000 | 0.015 | 0.031 | 0.050 | 0.071 | 0.090 | 0.105 | 0.115 | 0.122 | 0.124 |

---

> ### Author Response · Authors · 2025-11-21
>
> > ## **S3. Scaling to larger backbones**
>
> To evaluate whether AdvMask scales to larger architectures, we provided experiments on CLIP ViT-B/16 and ViT-L/14 in Appendix Sec. C.3. As shown in Table 10 below, AdvMask **continues to achieve substantial robustness gains even with those larger backbones.** These results demonstrate that AdvMask generalizes well across backbone capacities and remains effective on high-capacity models.
>
> - **Table 10. Evaluation on larger backbones**
> : Using ViT-B/16 and ViT-L/14 as image encoder, we report 16-shot test accuracy averaged over 5 datasets with 3 random trials.
>
> | Backbone        | Method      | Clean Acc. | Adv. Acc. |
> |-----------------|-------------|------------|-----------|
> | **ViT-B/16**    | CLIP        | 60.0       | 1.5       |
> |                 | FAP         | 66.8       | 33.9      |
> |                 | **AdvMask** | **68.9**   | **50.3**  |
> | **ViT-L/14**    | CLIP        | 67.0       | 3.7       |
> |                 | FAP         | 76.7       | 35.4      |
> |                 | **AdvMask** | **80.4**   | **63.4**  |
>
> &nbsp;
>
> > ## **S4. Per-class performance analysis**
>
> We thank the reviewer for the insightful suggestion. Following the request, **we performed a per-class accuracy analysis to identify which categories benefit the most from AdvMask in terms of adversarial robustness.** To summarize the results, we report the top-5 classes showing the largest robustness improvements on the Caltech101 dataset in Table 11 below. **AdvMask yields substantial gains over CLIP, particularly for categories that are highly vulnerable to adversarial perturbations.** Several classes where CLIP collapses to very low accuracy (e.g., car side, cellphone, okapi, face, ferry) recover to 70-100% accuracy with AdvMask. These categories typically rely on high-frequency or texture-sensitive cues, which are severely degraded by adversarial noise; AdvMask suppresses unstable activations and preserves semantically meaningful features. This analysis confirms that AdvMask provides the largest benefits for adversarially brittle or texture-dependent categories while maintaining strong overall performance. The detailed results have been added to Appendix Sec. D.9.
>
> - **Table 11. Key classes showing largest adversarial accuracy improvements**
> :Representative per-class adversarial robustness improvements obtained by AdvMask. We perform the analysis on Caltech101 dataset. Detailed results are provided in Appendix Sec. D.9.
>
> | Class     | CLIP Adv Acc (%) | AdvMask Adv Acc (%) | Δ (Improvement, %) |
> | --------- | ------------ | --------------- | --------------- |
> | car_side  | 0.0        | 100           | **+100**      |
> | cellphone | 11.8        | 100           | **+88.2**      |
> | okapi     | 9.1        | 100           | **+90.9**      |
> | face      | 1.5        | 95.4           | **+93.9**      |
> | ferry     | 5.0        | 95.0           | **+90.0**      |

---

> ### Comment · Reviewer_RUtB · 2025-11-24
>
> I thank the authors for conducting the additional experiments, and I appreciate their efforts. All my concerns have now been addressed. I still find the idea of masking parameters vulnerable to adversarial perturbations to be a fresh and valuable perspective on adversarial defense.

---

> > ### Author Response · Authors · 2025-11-25
> >
> > We are sincerely glad to hear that all concerns have been addressed. The reviewer’s suggestions were extremely helpful in improving our work. Once again, we thank you for the thorough review and for acknowledging our contributions.

---

### Official Review · Reviewer_LtP8 · 2025-11-01

**Soundness:** 3
**Presentation:** 3
**Contribution:** 2
**Rating:** 2
**Confidence:** 5

**Summary:**

This paper proposes an adversarial mask tuning (AdvMask) approach that searches for robust subnetwork within well-trained VLMs as a promising alternative which is highly parameter-efficient and is trained with Layer-wise Adaptive Feature Alignment (LAFA) loss. Experiments across various downstream datasets demonstrate that AdvMask consistently improves few-shot adversarial robustness over existing baselines.

**Strengths:**

The paper addresses an important problem of improving adversarial robustness in vision-language models (VLMs) with few-shot learning. The motivation is well-explained, and the paper is well-organized and clearly written. In figure 2, the authors provide a clear illustration of the proposed AdvMask method with different shots, which helps a lot in understanding the performance of the approach.

**Weaknesses:**

1. The proposed method is strongly related to adversarial model pruning, which has been extensively studied in the literature (arxiv.org/pdf/2409.01249). Therefore, the novelty of the proposed method is limited. The authors should also consider comparing with these adversarial pruning methods.

2. The improvement of the proposed method is not significant enough. From Table 4, adding AdvMask only improves around 0.7% robust accuracy and 1.5 % clean accuracy compared to directly using typical adversarial training.

3. The experimental settings are not strong enough. The authors should evaluate the performance of the proposed method with more adversarial settings, e.g., epsilon=2/155 or 4/255. See questions.

**Questions:**

1. Why is this method particularly effective for few-shot learning? What if using more training data?

2. Since the model parameters are changed after masking, does the author adopt adaptive attacks (e.g., PGD attack with knowledge of the mask) during evaluation? If not, the evaluation is not fair.

3. Can this method be better than existing zero-shot robust methods? e.g., simply adversarially train the entire model with one specific dataset like ImageNet?

4. Does this mask have any interpretability? For example, are they similar for different datasets?

---

> ### Author Response · Authors · 2025-11-21
>
> Dear Reviewer ```LtP8```,
> We sincerely appreciate your detailed and insightful review. The reviewer's comments are extremely helpful, and we carefully examined each point to provide thorough responses below. If you have any additional questions or further suggestions, we would be very grateful for the opportunity to clarify or elaborate.
> &nbsp;
>
> > ## **W1. Relation to adversarial pruning**
>
> We appreciate the reviewer for pointing out relevant work on adversarial pruning (AP). However, we respectfully clarify that AdvMask is fundamentally different from AP in its objective, optimization strategy, and problem setting.
>
> (1) **Objective.** AP methods [1-3] are pruning-based compression frameworks: they **permanently remove weights and then perform retraining or fine-tuning to recover accuracy**. In contrast, AdvMask is not a pruning or sparsity-driven method. Our goal is robust adaptation of pretrained VLMs in few-shot settings, not model compression. All pretrained weights remain untouched.
>
> (2) **Optimization target.** AP jointly optimizes weights (or weights + pruning masks) to obtain a sparse and robust model, followed by a retraining stage. In contrast, **AdvMask freezes all pretrained weights and learns only binary masks**, guided by the proposed LAFA loss to suppress layer-wise noise amplification. The model is never retrained after masking. As a result, AP incurs computational overhead due to retraining, whereas AdvMask performs parameter-efficient adaptation without retraining the pruned structure, and is specifically designed for few-shot VLM tuning.
>
> (3) **Problem setting.** To the best of our knowledge, this is the first work to introduce **mask-based robust adaptation for pretrained VLMs** in few-shot settings, where only mask parameters are optimized to discover robust neural pathways. This setting is not addressed in prior AP literature, which mainly focuses on supervised DNN trained from scratch, rather than large-scale pretrained VLMs requiring robustness recovery under extremely limited data.
>
> For these reasons, AP and AdvMask solve fundamentally different problems. We appreciate the reviewer’s suggestion and added a discussion of adversarial pruning in the related work section (Sec. 4) of the revised paper to clarify these distinctions.
>
> [1] Piras et al., Adversarial pruning: A survey and benchmark of pruning methods for adversarial robustness. PR'25.
> [2] Sehwag et al., HYDRA: Pruning Adversarially Robust Neural Networks. NeurIPS’20.
> [3] Chen et al., Sparsity Winning Twice: Better Robust Generalization from More Efficient Training. ICLR’22.
>
> &nbsp;
>
> > ## **W2. Performance improvement in loss ablation study**
>
> Thank you for pointing this out. We would like to clarify that the results in Table 4 correspond to a loss ablation study within our AdvMask. The row “$\mathcal{L}_{\text{CE+adv}}$” represents mask tuning using adversarial cross-entropy only (the base loss used inside AdvMask) while our LAFA loss is an auxiliary feature-level consistency loss designed to further enhance robustness. This ablation demonstrates two important points: (1) adversarial CE loss is highly effective and essential for mask tuning, and (2) adding LAFA consistently improves both clean and robust accuracy over all alternative auxiliary losses (e.g., JS divergence, KL divergence).
>
> To further validate the effectiveness of our loss design, we additionally include results under 1-shot and 4-shot settings in the revised version of Table 4. A summary is shown in Table 1 below. These results highlight that **the our loss design's improvement becomes more pronounced under low-shot conditions, where training data is scarce and feature representations are more susceptible to adversarial distortions.** In such scenarios, LAFA provides a stronger and more stable feature-level regularization signal than output-space divergence losses, enabling AdvMask to maintain coherent representations and achieve higher robustness. These additional experiments confirm that our loss formulation yields substantial robustness gains, especially when data is limited. We incorporated this clarification into the revised manuscript (in lines 431-436).
>
> - **Table 1. Additional loss ablation study**
> : We evaluate the performance of different auxilary losses under 1-, 4-, and 16-shot settings. Full results are provided in Table 4 of the main paper.
>
> | Method | Clean (1-shot) | Adv (1-shot) | Clean (4-shot) | Adv (4-shot) | Clean (16-shot) | Adv (16-shot) |
> | --- | -- | -- | --- | -- | --- | --- |
> | CLIP                                    | 56.6           | 4.8          | 56.6           | 4.8          | 56.6            | 4.8           |
> | $\mathcal{L}_{\text{CE-adv}}$           | 40.3           | 15.6         | 55.2           | 30.6         | 65.8            | 46.4          |
> | **$\mathcal{L}_{\text{LAFA}}$ (ours)**                    | **46.6**       | **18.4**     | **57.2**       | **32.2**     | **67.3**        | **47.1**      |

---

> ### Author Response · Authors · 2025-11-21
>
> > ## **W3. Robustness under different perturbation bounds**
>
> We agree that evaluating under stronger perturbations is important to support our claims. To address this, we have evaluated AdvMask under stronger adversarial settings, including higher perturbation bounds, as presented in Appendix C.1. As shown in Fig. 5 (and Table 2 below), **AdvMask consistently maintains strong adversarial robustness across multiple perturbation levels ($\epsilon$ = 1/255, 2/255, 4/255), demonstrating that its effectiveness is not limited to low-intensity attacks**. These results confirm that AdvMask provides scalable and reliable robustness even against stronger adversarial attacks.
>
> - **Table 2. Robustness under Different Perturbation Bounds**
> : We evaluate the robustness of different perturbation bounds ($\epsilon$ = 1/255, 2/255, 4/255). Experimental details are provided in Sec. C.1 of Appendix.
>
> | Adv. Acc.      | $\epsilon$=1/255            | $\epsilon$=2/255            | $\epsilon$=4/255           |
> | ----------- | -------------- | -------------- | ------------- |
> | **CLIP**    | 2.66           | 0.18           | 0.01          |
> | **FAP**     | 35.78 (± 1.10) | 11.08 (± 0.33) | 1.57 (± 0.32) |
> | **AdvMask** | **39.95 (± 0.22)** | **23.22 (± 0.14)** | **7.00 (± 0.20)** |
>
> &nbsp;
>
> > ## **Q1. Effectiveness of AdvMask in few-shot setting**
>
> Thank you for this important question. AdvMask is particularly effective in few-shot settings for several reasons. Prior work [1] has shown that optimizing binary masks (rather than model weight itself) is an efficient way to explore task-specific structure when only a limited number of samples are available. **By freezing all pretrained weights and learning only binary masks that activate or suppress specific pathways**, AdvMask preserves the strong generalizable representations of the pretrained VLM while selectively deactivating noise-sensitive or perturbation-vulnerable parameters. Moreover, our proposed LAFA loss provides a more stable and informative learning signal in few-shot regimes by enforcing feature-level consistency between clean and adversarial representations, while adaptive weighting encourages the model to focus on reliable samples and mitigates noisy overfitting that becomes particularly severe under data scarcity.
>
> When more training data are available, AdvMask continues to benefit from the additional examples. **As shown in Fig. 2-3, both clean and adversarial performance steadily improve as the number of tuning samples increases.** Other baselines also gain performance with more data, but AdvMask remains complementary as a lightweight parameter-efficient tuning method that enhances robustness with minimal cost. We clarified these points in the revised version of the paper (lines 347-350).
>
> [1] Zheng et al., Regularized mask tuning: Uncovering hidden knowledge in pre-trained vision-language models. ICCV'23.
>
> &nbsp;
>
> > ## **Q2. Mask-aware adaptive attack during evaluation**
>
> As the reviewer correctly pointed out, all evaluations in our paper **employ adaptive PGD attacks that have full knowledge to the learned mask.** Specifically, adversarial perturbations are computed by backpropagating through the masked model $f_{\theta
> \odot M_{bin}}$, ensuring a fully adaptive and fair evaluation setting. The strong robustness we observe under this adaptive attack confirms that our AdvMask genuinely suppresses vulnerable parameters and induces intrinsically robust feature representations, rather than relying on gradient obfuscation. We clarified this experimental setting explicitly in the revised paper (lines 312-314).

---

> ### Author Response · Authors · 2025-11-21
>
> > ## **Q3. Comparison with zero-shot robustness methods**
>
> We appreciate the reviewer’s thoughtful question. While our primary goal is to enhance adversarial robustness on a given downstream task using only a few examples, we also observe that **the learned masks exhibit strong zero-shot robustness**. To support this claim, we have included the results in Table 2 of the main paper (also provided in Table 3 below). Following the recently proposed zero-shot robustness method of TGA-ZSR [1], we first train each method using a held-out source dataset (i.e., TinyImageNet) and then evaluate zero-shot robustness on unseen target datasets without any further tuning.
> The results show that, although TGA-ZSR (entire) is trained on the full source dataset, our AdvMask trained on only 3.2% of the same dataset (i.e., 16 shots) achieves higher clean accuracy and only slightly lower adversarial robustness. Moreover, in the 16-shot setting, AdvMask achieves superior performance on both clean and adversarial samples compared to other baselines, demonstrating strong generalization from a very small amount of source data. Naturally, when few-shot samples from each downstream dataset are available (as in our main setting), AdvMask significantly outperforms TGA-ZSR, as shown in Fig. 2-3.
>
> These findings suggest that **our adversarial mask tuning strategy is not overly biased toward the training samples, but instead suppresses globally vulnerable parameters (i.e., invariant to dataset-specific distributions).** This allows the learned mask to generalize well to new, unseen environments, providing both few-shot and zero-shot robustness.
>
> - **Table 3. Results on zero-shot adversarial robustness (cross-dataset evaluation)**
> : Following [1], we first adapt the model using source dataset (i.e., TinyImageNet) and evaluate it on unseen downstream datasets. We report averaged clean and adversarial accuracy (%, ↑) over 5 datasets across 3 random trials.
>
> |           | shots (ratio)   | Clean Acc. | Adv. Acc. |
> | --------- | --------------- | ---------- | --------- |
> | CLIP      |                 | 61.9       | 2.7       |
> | TGA-ZSR   | entire (100%)   | 38.6       | 22.9      |
> |-----------|-----------------|------------|-----------|
> | FAP       | 16-shots (3.2%) | 36.0       | 16.8      |
> | TGA-ZSR   | 16-shots (3.2%) | 41.3       | 13.0      |
> | **AdvMask** | 16-shots (3.2%) | **42.0**       | **19.4**      |
>
> [1] Yu et al., Text-guided attention is all you need for zero-shot robustness in vision-language models. NeurIPS'24.

---

> ### Author Response · Authors · 2025-11-21
>
> > ## **Q4. Mask interpretability**
>
> Thank you for the constructive suggestion. Regarding mask interpretability, we provide a **layer-wise CKA analysis in Fig. 4 of Sec. 3.3, showing how the learned mask shapes a more stable and robust neural pathway.** Specifically, while pre-trained CLIP exhibits high similarity between clean and adversarial representations only in early layers (L1-L4), AdvMask preserves substantially higher similarity across all layers, effectively suppressing adversarial noise amplification and stabilizing the representations.
>
> Furthermore, as suggested by the reviewer, **we additionally conducted a dataset-level mask similarity analysis** during the rebuttal period. In particular, we computed pairwise IoU over the deactivated (masked) parameters across different datasets, and present the results in Fig. 8 of Appendix Sec. D.6. Also, a summary of the layer-wise IoU values is provided in the table 4 below.
>
> - **Table 4. Layer-wise IoU over masked positions (%)**
> : Layer-wise IoU over deactivated (i.e., mask=0) parameters for the learned masks under the 16-shot setting. The table reports Caltech101 vs. other datasets as examples. Results for other dataset pairs are provided in Appendix Sec. D.6.
>
> | Dataset Pair                 | L1   | L2   | L3   | L4   | L5   | L6  | L7  | L8  | L9  | L10 | L11 | L12 | Avg. |
> | ---------------------------- | ---- | ---- | ---- | ---- | ---- | --- | --- | --- | --- | --- | --- | --- | ---- |
> | Caltech101 vs. DTD           | 21.0 | 20.8 | 13.9 | 12.7 | 8.5  | 6.8 | 4.2 | 4.4 | 2.4 | 3.3 | 3.8 | 5.2 | 9.4  |
> | Caltech101 vs. FGVCAircraft  | 20.3 | 22.7 | 16.5 | 12.2 | 9.8  | 7.9 | 5.9 | 5.1 | 4.1 | 3.2 | 2.6 | 3.8 | 10.4 |
> | Caltech101 vs. OxfordFlowers | 22.1 | 25.4 | 20.9 | 16.2 | 11.7 | 8.2 | 6.2 | 5.6 | 4.5 | 3.6 | 1.9 | 4.0 | 11.5 |
> | Caltech101 vs. UCF101        | 25.4 | 26.7 | 20.7 | 16.4 | 11.5 | 8.8 | 6.5 | 6.0 | 4.4 | 3.8 | 2.7 | 4.3 | 12.4 |
>
> These results reveal two key findings: (1) Global similarity is relatively moderate (7.5%-12.4% on average), indicating that each dataset highlights somewhat different vulnerable parameter subsets, and (2) Early layers consistently exhibit higher IoU across datasets, suggesting that low-level feature extractors contain universally vulnerable parameters that AdvMask reliably suppresses. In contrast, later layers show more divergence, reflecting dataset-specific semantic adaptation and task-level behavior.
>
> In addition, to provide a more comprehensive interpretation, **Appendix Sec. D.6** includes further analyses on: Mask similarity across different runs (random seeds), Which layers or attention heads are primarily masked, and Which module types (e.g., Q/K/V projections vs. output projection) are most frequently masked. These expanded results offer a thorough and interpretable characterization of the learned masks and the structural vulnerabilities they capture.

---

### Meta-Review · Area_Chair_d5WS · 2026-01-05

**Summary:**

This paper proposes AdvMask, a parameter-efficient adversarial mask tuning method for pretrained vision-language models (VLMs) in few-shot scenarios. By learning binary masks to deactivate perturbation-vulnerable parameters and leveraging a layer-wise adaptive feature alignment (LAFA) loss, AdvMask improves adversarial robustness without modifying pretrained weights.

Reviewers had concerns regarding novelty, robustness evaluation, and clarity. Some reviewers (LtP8, kPP8) questioned novelty relative to adversarial pruning and single-modal fine-tuning methods. Others (RUtB, SmjY) requested stronger attacks, adaptive evaluation, and better differentiation between few-shot and zero-shot experiments. Authors addressed these by: (1) clarifying the distinction of AdvMask from prior works, (2) conducting stronger/adaptive attacks (ε = 2/255, 4/255, multimodal attacks), (3) providing visualizations and analyses of mask sparsity and interpretability, and (4) adding full fine-tuning and ablation baselines.

In my view, this paper makes a meaningful contribution to robust VLM adaptation under limited-data scenarios, provides thorough experimental validation. I recommend this paper for acceptance.

**Reviewer Concerns:**

**Addressed:**

Robustness under stronger/adaptive attacks (LtP8, RUtB, kPP8): experiments with higher ε and fully adaptive/multimodal attacks confirm stability.

Clarification of few-shot vs zero-shot settings (SmjY): definitions and table captions updated, performance differences explained.

Mask interpretability and visualization (RUtB, SmjY): layer-wise and head-wise sparsity, and visualizations provided.

Baselines and ablations: LAFA loss, and sparsity analyses show AdvMask is efficient.

Generalization to multimodal tasks (kPP8): additional image-captioning experiments demonstrate plug-and-play applicability.

**Outstanding:**

Limited gains on fine-grained datasets (RUtB).

Mask design and multi-modal coverage (kPP8) not fully explored, e.g., soft masks or text-branch masking.

**Reviewer Scores:**

LtP8: Likely would see a minor score increase, but still below acceptance.

RUtB: Likely to maintain a positive score, as stronger/adaptive attack results, mask visualizations, and full fine-tuning baselines address most concerns.

kPP8: Likely to see a slight score increase, as novelty and multi-modal masking questions are partially addressed.

SmjY: Likely to maintain a positive score, since clarifications on few-shot vs zero-shot, training efficiency, and mask visualizations resolve prior confusions.

---

### Decision · Program_Chairs · 2026-01-26

Accept (Poster)